# Effects of Phytochemicals from Fermented Food Sources in Alzheimer’s Disease In Vivo Experimental Models: A Systematic Review

**DOI:** 10.3390/foods12112102

**Published:** 2023-05-23

**Authors:** Alina Mihaela Baciu, Razvan Vlad Opris, Gabriela Adriana Filip, Adrian Florea

**Affiliations:** 1Department of Cell & Molecular Biology, “Iuliu Hatieganu” University of Medicine & Pharmacy, 6 Louis Pasteur Street, 400349 Cluj-Napoca, Romania; alinabacium@gmail.com (A.M.B.); adrian_a_florea@yahoo.com (A.F.); 2Department of Microbiology, “Iuliu Hatieganu” University of Medicine & Pharmacy, 6 Louis Pasteur Street, 400349 Cluj-Napoca, Romania; 3Department of Physiology, “Iuliu Hatieganu” University of Medicine & Pharmacy, 1-3 Clinicilor Street, 400006 Cluj-Napoca, Romania; gabriela.filip@umfcluj.ro

**Keywords:** Alzheimer’s Disease, fermented foods, phytochemicals, soy isoflavones, ginsenosides, kimchi phytochemicals, neuroprotection, cognitive function

## Abstract

The socioeconomic burden of Alzheimer’s Disease (AD) stems from its characteristic multifactorial etiology and, implicitly, the difficulties associated with its treatment. With the increase in life expectancy and health awareness, nutraceuticals and functional foods are filling in the gaps left by the limitation of classical medical treatment in chronic conditions associated with lifestyle factors, such as neurological disorders. Processes, such as fermentation that enhance food phytochemical content are garnering increased attention due to their functional and health-related properties. This systematic review aims to provide an overview of the evidence of phytochemicals from fermented food sources inducing therapeutic outcomes and cognitive benefits from in vivo experimental models of Alzheimer’s Disease. The present systematic review was conducted in accordance with PRISMA guidelines. Searches were performed in the following databases: MEDLINE, Embase, Cochrane, Scopus, Google Scholar, and Science Citation Index Expanded (Web of Science) by two independent reviewers. Titles and abstracts yielded by the search were screened for eligibility against the inclusion criteria. The search strategy yielded 1899 titles, encompassing studies from 1948 to 2022. After the removal of duplicates, and screening of titles, abstracts, and full texts, thirty three studies obtained from the original search strategy and seven studies from references satisfied the inclusion criteria and were included in the present systematic review. Several studies have emphasized the potential of fermentation to yield small-molecule phytochemicals that are not present in raw products. When these phytochemicals are combined, their collective strength has demonstrated the ability to exceed the antioxidant, anti-inflammatory, and neuroprotective benefits of individual phytochemicals when given in their pure form. Among the various fermented foods that have been studied, soy isoflavones obtained through fermentation have shown the most substantial evidence of altering phytochemical content and improving outcomes in animal models of AD. While promising in initial results, other fermented foods and traditional medicines require more detailed research in order to establish their effectiveness and proper utilization. As is, many of the experimental designs lacked phytochemical analysis of the used fermented product or comparison with the non-fermented counterpart. This, coupled with proper reporting in animal studies, will significantly raise the quality of performed studies as well as the weight of obtained results.

## 1. Introduction

Worldwide, approximately 55 million people live with dementia, of which it is estimated that Alzheimer’s Disease (AD) and Alzheimer’s Disease-related dementia account for approximately 70% of cases [1]. Projections estimate that the number of patients diagnosed with AD will double every 20 years [2]. In the United States, AD is, without doubt, the most widespread aging-associated neurodegenerative disease, with current estimations approximating the prevalence at 5.3 million cases. By the year 2050, this number is expected to rise to an alarming 11–16 million [3,4], while countries with middle and low income are also expected to be markedly affected. These figures highly suggest that lifestyle and environmental factors play an equally important role in the progression and onset of AD as the classically well-established factors (age, genetics, and socio-economic status). As life expectancy continues to grow, so does the elderly population, which in the following decades will undoubtedly cause the global healthcare system to face major demographic-induced fiscal pressure.

The social-economic burden of AD stems from its characteristic multifactorial etiology and, implicitly, the difficulties associated with its treatment. An ideal treatment for AD would need to target its multiple pathologic and biological processes intertwined in its occurrence and progression, including environmental factors, genetic predispositions, education, age, and lifestyle. The neuropathological trademarks of AD consist of abundant accumulation of extracellular Amyloid-β (Aβ) plaques, cerebral amyloid angiopathy, and irregular intracellular aggregation of hyperphosphorylated Tau protein, accompanied by microglial cell activation and astrogliosis. These modifications lead to activation of the inflammatory pathway, mitochondrial dysfunction, energy depletion, and ultimately to, the characteristic loss of synaptic elements, neurons, and neuropils. The culmination of cellular modifications manifests itself in patients as progressive neurocognitive impairment, displayed through the five A’s of AD: Amnesia, Apraxia, Agnosia, Aphasia, and Anomia [5]. Current treatments for AD that are approved by the FDA include three acetylcholinesterase inhibitors: rivastigmine, donepezil, and galantamine [6], memantine (a noncompetitive N-methyl-D-aspartate antagonist) [7] and a fifth drug currently in Phase 4 confirmatory trial: Aducanumab (ADUHELM^®^), a monoclonal antibody directed against Amyloid-β [8]. However, these drugs are not able to prevent or slow the progression of AD, as they only alleviate symptoms of the disease [7]. Therefore, there remains an urgent need for a curative or disease-modifying agent.

In the past decade, alternative treatments for AD have received an increasing amount of attention. These therapies range from nutraceuticals, functional foods, herbal adjuvants, and dietary supplements to meditation, aromatherapy, and acupuncture. Recently, research into nutraceuticals has focused on specific bioactive compounds and the mechanisms through which they exert their beneficial effect in the prevention and, potentially, the treatment of numerous afflictions. With the increase in life expectancy and health awareness, market expansion in healthcare spending, nutraceuticals, and functional foods are filling in the gaps left by the limitation of classical medical treatment in chronic conditions associated with lifestyle factors, such as neurological disorders. This has resulted in numerous attempts to modify available food so as to achieve higher health benefits. Medicinal and edible plants that have a high content of bioactive compounds have been investigated for the production of functional foods due to the increased antioxidant and anti-inflammatory capacity of the phytochemicals in their composition [9]. Food phytochemicals are garnering increased attention due to their functional and health-related properties. The general expectation is that the development of natural antioxidants, such as flavonoids, proteins, and phenolic compounds contained within plants, will enhance the safety and efficiency of treatment for chronic illnesses. In the case of neurodegenerative diseases, regular intake of phytochemicals with anti-inflammatory and antioxidant properties has been shown to slow the progression of neurocognitive decline and improve cognitive and physical performance [10]. Previous studies have focused their efforts on developing effective methods of increasing the phytochemical contents of functional foods and medicinal plants [11]. Processing techniques that have been used to alter and increase the phytochemical composition in food products include enzymatic pre-treatment [12], fermentation [13], autoclaving pretreatment [14], pulsed electric fields treatment [15], ultrasound-assisted extraction [16], and sun-drying [17]. Fermentation is a biotechnological process that makes use of enzymes produced by microorganisms to improve the organoleptic characteristics and nutritional value of food by converting conjugated phenolic forms to free phenolic forms while also altering the compositions of protein fractions [13,18]. Thus, fermentation deserves attention due to its potential to enhance the phytochemical composition of functional foods.

In order to evaluate novel therapeutics and clarify the fundamental mechanisms of AD, experimental models are a critical resource [19]. Prior to any clinical trials on human patients, in vitro and in vivo models are employed. While in vitro models facilitate the study of pathological changes at a cellular level, in vivo models have been developed to simulate human pathological changes associated with the disease. The major hallmarks of AD can be reproduced in these animal models, allowing for an in-depth analysis of AD pathogenesis and insights into a novel compound’s effects, mechanism of action, and toxicity [19]. For AD studies, small mammalian models such as rats and mice are most often used due to these animals’ complex behavior and nervous system similarity to that of humans. AD-like states can be induced in these models using transgenic models and artificial induction of AD (mechanical/chemical). However, an exact replica of the human AD pathology using these methods has not been achieved. Nevertheless, they offer the opportunity to safely assess the effects of novel phytochemical compounds in the search for neuroprotective adjuvant treatments for AD. Although numerous in vivo studies that document a wide range of increased beneficial effects of phytochemicals from fermented foods in AD have been published, the heterogeneity of the research warrants an in-depth analysis of the elicited effects. The present work provides a systematic overview of the evidence of phytochemicals from fermented food sources inducing therapeutic outcomes and cognitive benefits in experimental animal models of Alzheimer’s Disease.

## 2. Materials and Methods

This systematic review was performed according to the Preferred Reporting Items for Systematic Reviews and Meta-Analyses (PRISMA) guidelines [20]. A search was performed in the following databases: MEDLINE, Embase, Cochrane, Scopus, Google Scholar, and Science Citation Index Expanded (Web of Science). The search was performed using the following keywords: “fermented foods Alzheimer’s disease”, “phytochemical Alzheimer’s disease”, “ferment phytochemical Alzheimer’s disease”, “ferment neurodegeneration”, “ferment neuroinflammation”, “fermented memory impairment”. Search restriction applied: animal model/animal experiment/in vivo study.

### 2.1. Search Strategy

Two independent reviewers (A.M.B. and R.V.O.) screened the titles and abstracts yielded by the search against the inclusion criteria. Full reports were obtained for all titles that appeared to meet the inclusion criteria or where there was uncertainty. The two review authors then screened the full-text reports and decided whether these met the inclusion criteria. Additional information from study authors was sought where it was necessary to resolve questions about eligibility. Where there was a disagreement between the two reviewers, a consensus was sought through discussion. When a consensus was not achieved, a third reviewer who supervised the research activity (A.F. or G.A.F.) aided in the final decision. Reasons for excluding studies were recorded. None of the review authors were blinded to the study authors, affiliations, or journal titles.

### 2.2. Inclusion and Exclusion Criteria

Types of studies: We include published in vivo experimental studies. There were no location or language restrictions, nor were studies excluded based on the date of publication.

Experimental subjects: only studies which used animal models were included. We did not restrict sample sizes.

Intervention: We considered all studies in which phytochemicals from a fermented food source were administered to experimental models of Alzheimer’s Disease and experimental models of mechanisms and symptoms involved in Alzheimer’s Disease (including memory impairment, neurodegeneration, neuroinflammation, neuron cell death). We excluded studies that focused solely on predisposing factors to Alzheimer’s Disease (hypertension, hyperglycemia, hyperlipidemia) and did not present any effects at the level of the brain. Studies that administered fermented foods were included. We excluded studies that did not perform fermentation or use fermented products in order to obtain the tested phytochemicals (studies that purchased the tested phytochemicals directly in pure form from a supplier and did not specify if phytochemical was obtained through fermentation).

Outcomes: To be included, a study was required to present results from at least one of the following categories of mechanisms involved in Alzheimer’s Disease: behavioral (behavioral/cognitive/memory changes in animals), neuroinflammation, neurodegeneration and/or neuroprotection, antioxidant imbalance and/or protection at the brain level. Studies that presented outcomes on predisposing factors of Alzheimer’s disease without any data on brain-related effects were excluded.

### 2.3. Data Extraction

A data extraction form was constructed, which two review authors (A.M.B. and R.V.O.) used to extract data from eligible studies. Citation abstracts and full-text articles were uploaded to the form, in duplicate, with screening questions for inclusion eligibility. Extracted data were compared, with any discrepancies resolved through discussion, or if necessary, a third reviewer who supervised research activity was sought (G.A.F. or A.F.). When information regarding any of the above was unclear, we contacted the authors of the reports to provide further details.

The following data were collected:Report: author, year, and source of publication.Experimental design and features: number of animals included, sampling mechanism, treatment assignment mechanism, length.Animal model: species, with/without genetic modification, age, gender, weight.Intervention: type, duration, dose, timing, and mode of delivery.Main Outcomes: phytochemical analysis, cognitive function, Amyloid-β deposition, AChE activity, Ach levels, oxidative stress status, neuroinflammation.Secondary Outcomes: hypertension, hyperglycemia, hyperlipidemia.

Following data extraction from eligible manuscripts, references of the selected papers were checked in order to identify any other potential study that did not emerge from the first search. The inclusion and exclusion criteria for the selection of articles from references were the same as those previously used. The search strategy, inclusion, and exclusion criteria followed PRISMA guidelines [21]. The risk of bias for individual studies was assessed at the outcome and study level with the aid of SYRCLE’s risk of bias tool for animal studies [22].

## 3. Results and Discussion

The search strategy yielded 1899 titles, encompassing studies from 1948 to 2022. After the removal of duplicates and screening of titles, abstracts, and full-texts, thirty three studies obtained from the original search strategy and seven studies from references satisfied the inclusion criteria and were included in the present systematic review. The selection process of the included studies is detailed in the PRISMA flow diagram [23] Figure 1.

Out of the 40 in vivo experimental studies included in the present systematic review, 15 (37.5%) used rats as the primary animal model, while 25 used (62.5%) mice. The experimental models used consisted of AD induction by intracerebroventricular (i.c.v.) infusion of Aβ (*n* = 7, representing 17.5% of studies included in the review), scopolamine-induced memory deficiency (*n* = 10; 25%), transgenic animal models for AD including Tg2576 mice (*n* = 3; 7.5%), 3xTg-AD (*n* = 2; 5%), SAMP8 mice (*n* = 2; 5%), 5XFAD mice (*n* = 1; 2.5%), colchicine i.c.v. infusion (*n* = 1; 2.5%), trimethyltin (TMT) administration (*n* = 1; 2.5%), intraperitoneal injection (i.p.) of aluminum chloride (*n* = 1; 2.5%), lipopolysaccharide-induced neuroinflammation (*n* = 2; 5%), fructose and streptozotocin-induced type 2 diabetes (*n* = 1; 2.5%), alloxan-induced pre-diabetes (*n* = 1; 2.5%), hyperlipidemic diet (*n* = 1; 2.5%), deoxycorticosterone acetate (DOCA) salt-induced hypertension (*n* = 1; 2.5%), and combination disease induction (*n* = 6; 15%). The fermentation processes employed by the included studies consisted of bacterial fermentation with *Lactobacillus* spp. (*n* = 7; 17.5%), *Bacillus* spp. (*n* = 2; 5%), *Pediococcus pentosaceus* (*n* = 2; 5%), fungal fermentation with *Saccharomyces* spp. (*n* = 3; 7.5%), *Monascus* spp. (*n* = 2, 5%), *Trichoderma reesei* (*n* = 1; 2.5%), combination fermentation with two or more microorganisms (*n* = 7; 17.5%), and submerged fermentation (*n* = 1; 2.5%). It must be noted that 15 (37.5%) studies did not present the fermentation technique employed in obtaining the fermented product. In most cases, the reason behind this was that the authors acquired a commercially available food product which they then proceeded to use in the experiment. A little over half of the studies (*n* = 21; 52.5%) did not perform a phytochemical analysis of the tested fermented product, while a comparison between the effects of the fermented product to the original, non-fermented product was conducted by 20 (50%) of the studies included. A summary of study characteristics and their main findings is presented in Table 1.

The risk of bias for individual studies was assessed at the outcome and study level with the aid of SYRCLE’s risk of bias tool for animal studies [22]. As per the instructions of the risk of bias assessment tool, we present the summary results of the risk of bias assessment. Selection bias included inquiries into three areas of the study: sequence generation, baseline characteristics, and allocation concealment. For sequence generation, 55% (*n* = 22) presented as Unclear and 45% (*n* = 18) as High risk of bias. For baseline characteristics, 5% (*n* = 2) of studies presented High, 35% (*n* = 14) Unclear, and 60% (*n* = 24) Low risk of bias. For allocation concealment, 15% (*n* = 6) of studies presented High, 80% (*n* = 32) Unclear, and 5% (*n* = 2) Low risk of bias. Performance bias was assessed through inquiries about random housing and blinding. For random housing, 7.5% (*n* = 3) of studies presented Unclear and 92.5% (*n* = 37) Low risk of bias. For blinding, 12.5% (n-5) of studies presented High and 87.5% (*n* = 35) Unclear risk of bias. Detection bias assessed random outcome assessment and blinding. For random outcome assessment, 2.5% (*n* = 1) of studies presented High, 92.5% (*n* = 37) Unclear, and 5% (*n* = 2) Low risk of bias. For outcome blinding, 7.5% (*n* = 3) of studies presented High, 80% (*n* = 32) Unclear, and 12.5% (*n* = 5) Low risk of bias. Attrition bias assessed the presence of incomplete outcome data, for which 40% (*n* = 16) of studies presented High, 5% (*n* = 2) Unclear, and 55% (*n* = 22) Low risk of bias. Finally, reporting bias was assessed based on selective outcome reporting, for which 2.5% (*n* = 1) of studies presented High, 2.5% (*n* = 1) Unclear, and 95% (*n* = 38) Low risk of bias. The risk of biased results for individual studies is presented in Table 2.

In the following paragraphs, data about the investigated phytochemicals from fermented foods and their effects on AD are presented and discussed.

### 3.1. Soy Isoflavones

The main constituents of soybean (*Glycine max* (L.) Merr., family Leguminosae) are represented by dietary fibers, proteins, soyasaponin glycosides, and isoflavones. Soyasaponin glycosides have been shown to exhibit phytoestrogenic [64], antilipidemic [65], and memory-enhancing effects [66]. Its phytochemicals, isoflavones, and soyasaponins, have been extensively employed in neurodegeneration research for their antioxidant [62], anti-inflammatory [67], and memory-modulation effects [68]. It has been demonstrated that upon oral administration, isoflavone glycosides genistin and daidzin are metabolized by the gut microbiota to genistein and daidzein, respectively [69]. A similar trend has been observed with soyasaponins Ab and I to their aglycones soyasapogenols A and B [70]. Both in vivo and in vitro studies have shown that some of these metabolites exhibited more potent effects than their original glycosides (genistein demonstrated higher anti-inflammatory effects than genistin [71]; soyasapogenol B also exhibited a more potent phytoestrogenic effect than soyasaponin I) [70]. The transformation of isoflavone and soyasaponin glycosides to their aglycones can also be achieved through fermentation [24]. Given the complex phytochemical composition of soybean, it has the potential to impact multiple pathogenesis pathways of Alzheimer’s disease. This is reflected in the multiple approaches present in the body of studies on soybeans and their effects on AD. The following fermented soybean products have been investigated in the studies included in the present systematic review: defatted soybean powder, soybean, soymilk, Cheonggukjang, Doenjang, and Tempeh. Cheonggukjang and Doenjang are traditional Korean foods made by fermenting soybeans. While the fermenting processes used for these two products are similar, Cheonggukjang requires only a short-term fermentation with *Bacillus subtilis* [29], while Doenjang is obtained through a longer fermentation process and the use of salt brine and *Aspergillus* spp. [31]. Tempeh is a traditional Indonesian food obtained through short-term fungal fermentation (most often *Rhizopus oryzae* or *Rhizopus oligosporus*) of soybeans [32].

In the first study conducted by Yoo et al. (2015) [24], the effects of defatted soybean (*Glycine max* (L.) Merr.) powder fermented with *Lactobacillus pentosus* (var. plantarum C29) on a mouse model (male, ICR, 6 weeks old) with scopolamine-induced memory impairment (intraperitoneal injection 1 mg/kg b.w.) were evaluated. The results were further compared to those obtained from administering a non-fermented defatted soybean ethanol extract counterpart. The oral treatments consisted of soy powder ethanol extract (100 mg/kg), fermented soy powder ethanol extract (100 mg/kg), soy powder (500 mg/kg), fermented soy powder (500 mg/kg), and tacrine (10 mg/kg). The products were administered 30 min before the scopolamine injection, after which the mice were subjected to the passive avoidance test (PAT), Y-maze, and Morris Water maze (MWM). The authors found that fermentation increased the soybean content of genistein, soyasapogenol A, soyasapogenol B, and daidzein. This, in turn, was correlated with increased expression of hippocampal brain-derived neurotrophic factor (BDNF) and attenuated acetylcholinesterase (AChE) activity compared to the non-fermented soybean powder. The effects were maintained in the behavioral tests, where the authors reported efficient mitigation of scopolamine-induced memory impairment in mice that received soybean powder and fermented soybean powder, with the latter expressing more prominent results comparable to the positive control (tacrine).

In another study by Lee et al. (2018) [25], *Lactobacillus plantarum* C29-fermented soybean (*Glycine max* (L.) Merr.) (100 mg/kg b.w./day, suspended in 1% dextrose) and the probiotic alone (1 × 10^9^ CFU/mouse) were administered to 5XFAD transgenic mice (4 months old) for a period of 2 months (by gavage, daily). These transgenic mice are often used in animal models of AD due to the significant accumulation of Aβ plaque at the brain level, which manifests itself as impaired cognitive function by the time the animals reach 4 months of age. Following the administration of treatment, the authors proceeded with performing the PAT, novel object recognition test (NORT), and MWM. While control mice that had not received the test treatments exhibited significantly impaired cognitive function compared to normal mice, both treatments alleviated the decrease in cognitive function. Furthermore, both treatments inhibited hippocampal nuclear factor kappa-light-chain-enhancer of activated B cells (NF-kB) activation while concurrently increasing BDNF expression and cyclic adenosine monophosphate (cAMP) response element binding protein (CREB) phosphorylation. A noted suppression of Aβ plaque accumulation at the cortex and hippocampus level was also observed. Treatments also significantly suppressed β-secretase-1 (BACE1), Aβ, Psen-1 (γ-secretase 1), and caspase-3 expression in the cortex and hippocampus of the transgenic mice. This result was reinforced by confocal microscope-mediated identification of apoptotic neuronal cell populations, which were suppressed in treated mice. It is important to note that while both treatments revealed statistically significant results, those observed for the fermented soybean were increased compared to the *Lactobacillus plantarum* probiotic treatment. These results hint that the fermented soybean product can suppress Aβ expression and Aβ-induced neuronal cell death through NF-κB activation and β-secretase, γ-secretase, and caspase-3 expression inhibition. The treatments also affected the intestinal flora, where a decrease in *Enterobacteriaceae* and an increase in lactobacilli/bifidobacterial populations were observed, essentially restoring the altered intestinal flora present in transgenic mice. This, coupled with suppressed fecal and blood lipopolysaccharide (LPS) levels and reduced NF-kB activation, permit another possible mechanism of action, modulation of gastro-intestinal inflammation. In a second study by the same group of researchers, Lee et al. (2017) [26] focused on this second potential mechanism. The animal model used was ICR mice (6 weeks old) with LPS-induced memory deficit (intraperitoneal injection of LPS 8 μg/kg b.w./day for 10 days) treated with *Lactobacillus plantarum* C29-fermented defatted soybean (*Glycine max* (L.) Merr.) (80 mg/kg b.w.), defatted soybean (80 mg/kg), and its constituents soyasaponin I, soyasapogenol B, genistein and genistin (each 10 μM). All treatments inhibited LPS-induced phosphorylation of NF-κB signaling molecules IκBα and TAK1 (*p* < 0.05) as well as activation of NF-κB (*p* < 0.05). Soyasapogenol B showed the most pronounced inhibition of NF-κB activation, followed by genistein. It is important to note that the inhibitory effect of fermented defatted soybean was stronger than that of simple defatted soybean. Interestingly, Soyasapogenol B and genistein increased ERK phosphorylation and BDNF expression more efficiently compared to their glycoside counterparts. Effects on mice cognitive impairment were tested with the aid of behavioral tests, in which the fermented soybean treated group performed better than the C29 probiotic group, with significant recovery from the LPS-induced reduction in spontaneous alterations (Y-maze) and decreased latency time (PAT). However, no difference was observed in mice that had not been previously treated with LPS. The same pattern was observed with the constituents of the fermented soybean, which also efficiently induced recovery from LPS-induced reduction in spontaneous alterations (soyasaponin I 93.0%, genistin 96.8%, genistein 101.0%, soyasapogenol B 105.7%), with no effects on mice that were not treated with LPS.

A different approach was taken by Liu et al. (2016) [27], which highlighted the potential for soymilk fermented with *Lactobacillus plantarum* strain TWK10 to mediate neuronal cell degeneration present in AD through a decrease in oxidative stress damage and high blood pressure. The experimental model consisted of Wistar rats (8 weeks old) with vascular dementia induced by the administration of deoxycorticosterone acetate (DOCA) (subcutaneously 20 mg/kg b.w. twice per week for 90 days) and a salt solution (1% NaCl and 0.2% KCl) starting from day 85 of the experiment. The groups that had received fermented soymilk water (2.65 g/kg b.w. for 5 weeks) and fermented soymilk ethanol extract (0.09 g/kg b.w. for 5 weeks) presented inhibited AChE activity (*p* < 0.05), suppressed MDA serum levels, increased nitric oxide (NO) production, glutathione (GSH) level (*p* < 0.05), catalase (CAT) activity (*p* < 0.05), and decreased blood pressure (by 11.23% and 14.50%, respectively). The two test groups presented significantly restored superoxide dismutase (SOD) activity (*p* < 0.05) that had been reduced by the DOCA experimental model. These results were correlated with decreased escape latency and increased target crossing (*p* < 0.05) in the MWM. The observed modulation of oxidative stress, AChE activity, and improved cognitive function perpetuate the idea that fermented soybean products of numerous forms could be employed for the alleviation of AD hallmarks. Inhibition of AChE activity has also been achieved by Go et al. (2016) [28] by administering Cheonggukjang, a traditional Korean fermented soybean product that was obtained by fermentation with a mixed culture of *Lactobacillus sakei* 383 and *Bacillus subtilis* MC31. ICR mice (6 weeks old) were pre-treated for 4 weeks with three different doses of Cheonggukjang (25 mg, 50 mg, and 100 mg/kg b.w., orally) followed by intraperitoneal injection of a single dose of trimethyltin chloride (2.5 mg TMT/kg body weight). In addition to inhibition of AChE activity, treatment groups registered a dose-dependent increase in the concentration of nerve growth factor (NGF), activation of the nerve growth factor receptor signaling pathway (TrkA and p75NTR receptors), and lower Bax/Bcl-2 levels. Significant oxidative stress modifications were observed in the form of enhanced SOD activity and steeply decreased levels of MDA (43–58% lower). Trimethyltin-induced long and short-term memory deficit was significantly improved in pretreated groups, as observed through the PAT and NORT behavioral tests. Lee et al. (2013) [29] also reported increased NGF concentration and phosphorylation levels of TrkA and Akt in their investigation of the effects of Cheonggukjang (oral administration for 8 weeks) on a mouse model (Tg2576, 15 weeks old, female) of AD. However, no modification was observed in the NGF receptor p75NTR signaling pathway between treatment and control groups despite a significant increase in NGF levels by the fermented product (*p* < 0.05).

Aside from the major pathologic modifications that are associated with AD, hyperglycemia is an important underlying mechanism of AD progression that must be considered in a world where type 2 diabetes and insulin resistance are quickly rising in incidence. Numerous epidemiological studies conducted over the past 30 years have shown that individuals with type 2 diabetes mellitus have an increased risk of dementia (between 50 and 150%) compared to the general population [72,73]. Recent evidence has suggested the existence of a reciprocal causative link between impaired glucose regulation and early-stage Aβ plaque deposition [74]. In patients with type 2 diabetes, hyperglycemia may lead to neuron damage through oxidative stress, inflammation, osmotic irregularities, and protein deformation through advanced glycated end-products [75]. AD symptoms in patients with type 2 diabetes and hyperinsulinemia have also been shown to be accelerated through increased Aβ aggregation in the hippocampus [76]. On the other hand, glycemic control through the use of anti-diabetic medications has been shown to reduce the risk of AD [77]. To this end, Yang et al. (2015) [30] examined the potential of Chungkookjang fermented using both the traditional method and *Bacillus lichenifomis* to protect against glucose dysregulation and cognitive dysfunction in a rat model (male, Sprague–Dawley) of AD (hippocampal infusion of Aβ25–35 3.6 nmol/day for 14 days)) and type 2 diabetes (90% pancreatectomy). The treatments were administered orally for 8 weeks and consisted of 10% lyophilized cooked soybeans, traditional Chungkookjang, and Chungkookjang fermented with *Bacillus lichenifomis.* The *Bacillus* fermented Chungkookjang presented the highest content of isoflavonoid aglycones. Consequently, rats fed with this soy product presented the highest decrease in hippocampal Aβ deposition, Tau phosphorylation and expression, and improved insulin signaling (increased CREB, Akt, and GSK phosphorylation). The experimentally induced cognitive impairment was also markedly attenuated in this treatment group as measured by the PAT and MWM. *Bacillus* fermented Chungkookjang also normalized hepatic glucose output and whole-body glucose infusion rates during the euglycemic hyperinsulinemic clamp, along with restoring β-cell mass in AD rats. These results indicated that different fermentation techniques of Chungkookjang may improve brain glucose metabolism in the brain, thus also mitigating memory dysfunction in AD. Through a similar approach, the effects of Doenjang, a traditional Korean fermented soybean product, were investigated for its potential protective effects against neurodegeneration and neuroinflammation in a mouse (C57BL/6J mice) model of obesity (fed a high-fat diet for 11 weeks) by Ko et al. (2019) [31]. The high-fat diet increased neuronal loss (spectrin α breakdown products, C/EBP homologous protein, and cleaved caspase-3 protein levels), which was attenuated in the Doenjang treatment group. Likewise, mice treated with Doenjang presented higher mRNA levels of neurotrophic factor and protein cell nuclear antigen levels, effectively participating in the regulation of the CREB-BDNF pathway and neuron cell proliferation. The high-fat diet to which the animals were subjected led to increased expression of heme oxygenase 1 (*p* < 0.05), carbonylated protein (*p* < 0.05), and TBARS (*p* < 0.05) contents, all of which were attenuated by administration of Doenjang. At the level of the hippocampus and cortex, the treatment also significantly decreased diet-elevated mRNA levels of neuroinflammatory genes, including TNFα (*p* < 0.05), IL-6 (*p* < 0.05), caspase 1 (*p* < 0.05), monocyte chemoattractant protein 1 (*p* < 0.05), and Toll-like receptor 4 (*p* < 0.05). Effects were observed on Aβ deposition through regulation of gene expressions involved in Aβ production and degradation (downregulated mRNA levels of presenilin 1 (*p* < 0.05) and BACE1 (*p* < 0.05), and upregulated levels of insulin-degrading enzyme (*p* < 0.05). Finally, Doenjang consumption reduced tau hyperphosphorylation, and glycogen synthase kinase activity increased by the high-fat diet in the hippocampus and cortex of mice. In all investigations, the Doenjang-treated mice presented increased effects in comparison to those treated with steamed soybean. Ultimately, these results highly suggest that the phytochemical compounds produced by the fermentation of Doenjang may be effective at enhancing the neuroprotective effects of Doenjang.

Another fermented soybean food product investigated for potential benefic effects on Alzheimer’s Disease is Tempeh, a traditional food originating from Indonesia. In a study by Chan et al. (2018) [32], three different doses of Tempeh (300, 600, and 900 mg/kg b.w./day) were administered to a senescence-accelerated mouse model (SAMP8 mice, 6 months old) over a 12-week period. Noted results were observed for mice that were fed the highest dose of Tempeh, with higher SOD (*p* < 0.05) and CAT activity (*p* < 0.05) in the striatum and hippocampus coupled with increased expression of mRNA when compared to the control group. This high dose also influenced responses to oxidative stress by modulating the mitogen-activated protein kinase (MAPK) pathway, increasing expression of nuclear factor erythroid 2-related factor 2 (Nrf2), and downregulating p-JNK levels (*p* < 0.05) and p-p38 levels (*p* < 0.05). This dose also significantly reduced BACE1 (*p* < 0.05) and Aβ (*p* < 0.05) in SAMP8 mice to levels close to control. This cascade of effects culminated in improved cognitive function in mice, as demonstrated in passive avoidance time. A different approach was taken by Ayuningtyas et al. (2019) [33], where rats (Wistar, male) were injected with alloxan (single i.p., 120 mg/kg b.w.) to induce prediabetes and then treated with Tempeh (low dose of 9 g/200 g b.w. and high dose of 18 g/200 g b.w.) for a period of 14 days. Tempeh treatment was not able to significantly affect spatial memory (tested through MWM) or blood glucose levels. While the experimental conditions between these two studies differ significantly, it is interesting to note the large discrepancy between their results. Despite administering a much larger dose of Tempeh, no significant improvements were noted by Ayuningtyas et al. (2019) after 14 days of treatment. This brings into question the action mechanisms of the fermented foods, the duration until the inception of effects, and possible interferences caused by the experimental model. This last point must be stressed as alloxan is primarily used to induce type 1 diabetes in animals by completely suppressing islet response to glucose for a limited period of time [78]. Alloxan administration has been associated with an increase in brain monoamines [79], an imbalance in antioxidant enzymes, and an increase in anxiety-like behavior [80]. On the other hand, SAMP8 mice classically present age-associated behavioral impairments such as memory, learning difficulties, and reduced anxiety-like behavior [32].

Natto, a traditional Japanese fermented soybean dish, is rich in bioactive compounds that contribute to its potential therapeutic effects in addressing cognitive deficits associated with AD. These compounds include soy isoflavones, nattokinase, and menaquinone-7, each with distinct functions [81]. Soy isoflavones, represented by daidzin and genistin, possess antioxidant and anti-inflammatory properties, protecting brain cells from damage caused by oxidative stress and inflammation. They also exhibit estrogenic activity, influencing neurotransmitter systems and promoting neuronal health [69]. Natto is a notable source of nattokinase, an enzyme with fibrinolytic activity. By breaking down fibrin and promoting healthy blood circulation, nattokinase has been shown to support brain health and improve cerebral blood flow, which is essential for optimal cognitive function [82]. Menaquinone-7, a form of vitamin K2 found in natto, plays a crucial role in the brain by influencing the synthesis and metabolism of sphingolipids. These sphingolipids are responsible for regulating cognitive function through their involvement in various neuronal processes, including proliferation, differentiation, cellular communication, and aging [83]. Moreover, it is involved in the enzymatic activation of two important proteins that contribute to maintaining cerebral homeostasis. The first protein, Growth-arrest specific 6, exhibits anti-apoptotic, mitogenic, and myelinating properties and thus plays a vital role in protecting neurons and promoting their growth. The second protein is known as protein S, which demonstrates neuroprotective effects and helps maintain the integrity of the blood–brain barrier. The combination of these bioactive compounds in natto offers a synergistic effect, enhancing their individual functions and potential therapeutic benefits [84]. By incorporating natto or its bioactive components into nanonutraceutical (NN) formulations, it is possible to harness these bioactive functions effectively. These nanonutraceuticals can be designed to improve the bioavailability and targeted delivery of the bioactive compounds, ensuring their optimal therapeutic impact on cognitive function in AD. This avenue was explored by Bhatt et al. (2018) [34] in a recent study on a rat model (Wistar, male) of AD-like cognitive deficit (i.c.v. injection of colchicine 15 µg/5 µL). Following solid-state soybean fermentation with *Bacillus subtilis* MTCC 2616, daidzin (185.5 lg/mL), nattokinase (93.6 FU/mL), glycitin (81.85 µg/mL), genistin (117.71 µg/mL) and menaquinone-7 (110.77 µg/mL) were extracted and used to create NN with a particle size of 201.5 nm (±1.45 nm) and administered orally, daily (dose of 0.5 g/200 g b.w.) to the animals for a period of 32 days. The treatment reduced hippocampal activity of GSH (by 42%), CAT (41%), SOD (43%), carbonyl protein levels (30%), and lipid peroxidation (28%) while also increasing the activity of AChE by 42%. The NNs also demonstrated effective inhibition of BACE-1 and amyloid-β activity during in silico studies. These effects were associated with potent reversal of memory and learning impairment caused by the colchicine treatment observed in the PAT and the MWM.

The encouraging results of studies that have tackled fermented soybean products highlight its potential use as a multi-phytochemical compound that can simultaneously target multiple AD pathogenesis pathways. GE and SB showed a potent ability to improve memory and higher cognitive function through an increase in BDNF while concurrently preventing neuroinflammation through suppression of TNF-α expression and NF-κB activation in the hippocampus. Daidzin has the ability to interact with BACE proteins efficiently and bind to formed Aβ, thus both reducing production and facilitating degradation of Aβ. In addition to demonstrating potent antioxidant effects by affecting the CREB-BDNF pathway, soy isoflavones have also shown the ability to mitigate hyperinsulinemic states. By employing fermentation processes, these natural phytochemicals contained in soybeans have been shown to be increased, likely due to a combination of factors, including the creation of their more potent metabolites, probiotic effects of the microorganisms used in the fermentation processes, and creation of fermentation-specific enzymes that have the ability to impact Aβ formation and degradation (e.g., nattokinase).

### 3.2. Ginsenosides

Ginseng (*Panax ginseng* C.A. Meyer) is one of the most popular traditional herbal medicines in East Asia that has been thoroughly investigated for its numerous pharmacological effects, including its potential neuroprotective effects and mitigation of dementia-associated cognitive dysfunction [85]. Ginsenosides (triterpene saponins) represent the main active ingredients in ginseng. These have been classified into two categories: 20(S)-protopanaxadiol (PPD) (Rs1, Rb1, Rb2, Rb3, Rd, Rg3, Rc, Rh2) and 20(S)-protopanaxatriol (PPT) (Rf, Re, Rg1, Rh1, Rg2), with the difference between them consisting of a carboxyl group at the C-6 position in PPDs [86]. Each of these ginsenosides has been shown to present different effects. Upon ingestion, they are hydrolyzed by intestinal bacteria into their major active metabolites, 20(S)protopanaxatriol (M4) and 20(S) protopanaxadiol 20-O-beta-D-glucopyranoside (M1/compound K) before being absorbed [87]. This suggests that the effects of ginsenosides are dependent not only upon their concentration in the original root but also on the composition of an individual’s intestinal flora. Given that Alzheimer’s Disease is associated with altered microbiota [88] and intestinal inflammation [89], it is likely that the efficacy of ginsenosides is reduced in patients suffering from this form of dementia.

The following experimental studies employed bacterial fermentation in order to circumvent this predicament. Hence, Nagao et al. (2019) [35] investigated the effects of ginseng fermented with *Lactobacillus paracasei* A221 (dose of 100 and 300 mg/kg b.w, daily, orally for 7 days) versus non-fermented ginseng (300 mg/kg b.w) on an animal model (Wistar rats, 10 weeks old, male) of AD (7 days intracerebroventricular infusion of Aβ1-42; a dose of 600 pmol/20 µL per day) and cerebral ischemia (bilateral electrocauterization of the vertebral arteries). These fermented treatments were administered immediately after disease induction. Significant results were obtained in groups treated with the higher dose of fermented ginseng, including a decrease in hippocampal Iba-1, caspase-3, and cleaved caspase-3 protein expression, neuron loss reversal in the CA1 region of the hippocampus (recovery of neuronal nuclei-positive cells seen through immunohistochemistry), as well as spatial memory amelioration (MWM). Interestingly, although the ginseng used in this study contained a good quantity of ginsenosides, it did not present any neuroprotective effects at the administered dosage compared to its fermented counterpart. Similar results were obtained by An et al. (2019) [36] by analyzing the effects of wild ginseng root fermented with *Pediococcus pentosaceus* (doses of 150, 300, and 600 mg/kg b.w., orally, daily for 3 weeks) on an animal model (C57BL mice, male) with scopolamine-induced memory dysfunction (intraperitoneal injection of 1 mg/kg b.w. 30 min after treatment administration). The fermentation process successfully increased the Rg5 (21.48 mg/g) and Rk1 (18.71 mg/g) contents of the raw material, which were originally present only in trace amounts. The fermented product, as well as donepezil (the positive control), significantly inhibited AChE activity while preserving Ach levels in the animal brains. The animals treated with the fermented product also showed improvement in memory during MWM, Y-maze test, and PAT. In a similar study by Kim et al. (2021) [37], the authors obtained a significant alteration of ginsenoside content through fermentation of ginseng with *Pediococcus pentosaceus*, specifically small molecule ginsenosides Rg3 (44.26 mg/g), Rk1 (15.93 mg/g), and Rg5 (23.10 mg/g) were detected in the fermented product. The fermented product (125, 250, and 500 mg/kg b.w.) was administered to two experimental models. In the first, male mice with scopolamine-induced memory deficit (i.p. 1 mg/kg b.w. 30 min after administration of test agents) were used. In preparation for disease induction, the tested substances were administered once per day for a period of 7 weeks. Scopolamine was administered only on the days on which the behavioral tests were performed. In the second model, female mice were ovariectomized, and then D-galactose (s.c. 100 mg/kg b.w.) was administered once per day, 3 days/week for 6 weeks, followed by 7 days/week for 2 weeks, for a total of 8 weeks in order to induce aging. The tested fermented product was concurrently administered daily, orally, for a period of 8 weeks. Treatment with fermented ginseng in doses of 250 mg or higher downregulated AChE activity (*p* < 0.01) while restoring Ach levels (*p* < 0.05) in female mice groups. All treatment groups saw a lowering in MDA concentration and an increase in CAT activity. No data regarding AChE, ACh, MDA, or CAT were presented for the male mice groups. Behavioral testing through MWM revealed improved spatial memory of treatment groups (both female and male groups), while no statistically significant modifications were noted through the Y-maze test.

Fermented ginseng products demonstrated the ability to positively affect phytochemical content and oxidative stress, and improve cognitive performance while also exhibiting neuroprotective effects at the structural level.

### 3.3. Kimchi Phytochemicals

Kimchi, a traditional Korean fermented food made with red pepper, cabbage, garlic, green onion, ginger, and fermented fish sauce, has high contents of phytochemicals including quercetin, quercitrin, capsaicin, 3-(4′-hydroxyl-3′,5′-19 dimethoxyphenyl)propionic acid (HDMPPA), and ascorbic acid [90] that have been previously studied for their beneficial effects in AD. However, very few studies have compared the efficiency of the whole food product versus the individual phytochemicals in animal AD models. One such study by Woo et al. (2018) [38] performed a comparative study between the major phytochemical components in Kimchi and the aqueous extract of traditional fermented Kimchi on a mouse (ICR mice, 5-week-old) AD model (singular i.c.v. administration of Aβ25-35 5 nmol/5 μL). Over a 2-week period following the surgical intervention, mice were orally administered the different treatments in the following doses: capsaicin 10 mg, quercetin 50 mg, HDMPPA 50 mg, ascorbic acid 116 mg, and Kimchi aqueous extract 200 mg/kg b.w./day. All bioactive compounds and Kimchi treatments led to a decrease in BACE, APP, phosphorylated Tau, p-PERK, GRP78, XBP1, p-EIF2α, and CHOP protein expression (*p* < 0.05). Treatments increased levels of pro-apoptotic molecules cIAP and Bcl2, while p-JNK and Bax were decreased (*p* < 0.05), as well as cleaved caspase 3 and 9 (*p* < 0.05). Only HDMPPA, quercetin, and the Kimchi extract affected oxidative stress by reducing TBARS and ROS levels (*p* < 0.05). Following this, a TUNEL assay was performed to assess DNA fragmentation and cell apoptosis. The TUNEL assay supported the aforementioned results as the hippocampus of rats from treatment groups presented significantly fewer TUNEL-positive cells. In a continuation of this study [39], in which the authors tested the same phytochemicals and fermented food (same dosages), an increase in protein expression levels of antioxidant enzymes (Nrf2, SOD, and GSH) and a decrease in inflammation-related enzymes (NF-kb, cyclooxygenase-2, and inducible nitric oxide synthase) (*p* < 0.05) was reported in all treatment groups. Treated groups also performed better in the T-maze, MWM, and NORT. Notably, groups that received quercetin and Kimchi extract (200 mg/kg b.w.) presented the most promising results, although the individual concentrations administered to the animals were higher than those contained in the Kimchi product. Although there are limited studies on the connection between Kimchi and AD, the initial findings are highly promising and emphasize the potential of Kimchi’s diverse range of natural phytochemicals for neuroprotection. Given the well-established impact of quercetin on AD pathogenesis, these studies provide evidence that its effects can be further developed and amplified by incorporating probiotic-facilitated multi-phytochemical administration.

### 3.4. Alcoholic Beverages Phytochemicals

Wine is one of the most widely consumed alcoholic beverages in the world. Obtained through careful fermentation of grapes, the amount and potency of their phytochemicals (main: anthocyanins and resveratrol) highly depends on the type of grapes used in their creation. This, in turn, can signal that different wine assortments have the potential to enact different effects on the pathogenesis pathways of AD. Wang et al. (2006) [40] studied the effects of moderate consumption of Cabernet Sauvignon (grapes used were grown in Fresno, CA, USA) on Tg2576 mice (female, 4-month-old). The wine was administered daily in drinking water (6% final alcohol concentration) over a period of 7 months. Their results showed that the wine promoted non-amyloidogenic processing of amyloid precursor protein through a neocortical increase in alpha-secretase activity (*p* < 0.05), a higher concentration of soluble N-terminal fragment of APP (*p* < 0.05), and an increase in the concentration of membrane-bound alpha-CTF cleaved fragment (*p* < 0.05). These changes ultimately led to the prevention of Aβ peptide generation, as evidenced by the decrease in Aβ1–42 and Aβ1–40 peptides in the hippocampus (*p* < 0.05) and neocortex (*p* < 0.05). Through the Barnes maze test, the authors also ascertained the presence of a marked cognitive improvement in the treatment groups. Interestingly, the minimum concentration of resveratrol that has been shown to promote Aβ clearance in vitro is 10 times higher than the concentration obtained in the studied wine (0.2 mg/L of resveratrol). In a secondary study, Ho et al. (2009) [41] administered for a period of 10 months a muscadine wine (diluted in water to a 6% alcohol content) which also significantly impacted AD-type brain pathology and memory deterioration in a transgenic AD mouse model (Tg2576 mice, 4 months old). In stark contrast with the previous study, no detectable change in brain α-, β-, or γ-secretase activity was observed. Instead, the authors observed reduced levels of soluble high molecular weight oligomeric Aβ species in the hippocampus and cerebral cortex of treated mice coupled with a significant improvement in cognition and spatial memory (observed in the MWM test). These studies suggest the possibility of achieving an increased anti-AD effect by developing a concentrated combination of dietary polyphenolic compounds that have the possibility to simultaneously affect multiple Aβ-related mechanisms instead of relying on a single-phytochemical treatment approach. An example of such a strategy was elaborated by Mendes et al. (2018) [42] in an experimental study in which a polyphenol-enriched diet was administered (for 2 months, daily) to 3xTg-AD mice (10-month-old) by supplementing the drinking water with polyphenolic wine extract (100 mg/L gallic acid equivalents). The extract was obtained by polyvinylpyrrolidone polymer adsorption of white wine (Portugal origin, Douro Region). The polyphenols-enriched diet promoted brain accumulation of hydroxybenzoic acid derivatives and catechin and led to an increase in CAT activity and glutathione/glutathione disulfide ratio and a decrease in membrane lipid oxidation (TBARs level). The functional diet also decreased brain levels of Aβ1-42 and Aβ1-40. However, the treatment did not produce any significant attenuation of the brain mitochondrial bioenergetic dysfunction seen in the 3xTg-AD mice.

In addition to wine, other fermented drinks have shown potential benefits in AD. The most consumed alcoholic beverage in the world is beer [91]. Cecarini et al. (2022) [43] have examined the effect of unpasteurized and unfiltered beer (9% alcohol content) on 3xTg-AD mice (2 months old). The beer was enriched with *Saccharomyces cerevisiae* and administered orally daily (6/7 mL/day) for a period of 4 months. The enriched beer led to a marked decrease in Aβ1-42 in the hippocampus (*p* < 0.01) and prefrontal cortex (*p* > 0.05) of the transgenic mice, a decrease in TNFα (*p* < 0.05) and IL-1β (*p* < 0.01), coupled with a marked increase in IL-10 (*p* < 0.05) and IL-4 (*p* < 0.01) in both regions. Additionally, a marked modification in intestinal flora was noted with an important decrease in *Sordariomycetes* fungi, which have been previously associated with intestinal inflammatory conditions. The memory and learning abilities of the mice were tested with the use of the NORT, where the group treated with the yeast-enriched beer performed best.

Aside from popular alcoholic beverages, numerous other less-known drinks have the potential also to affect different mechanisms involved in AD. One such example is Raffia palm wine which was investigated by Erukainure et al. (2019) [44] for its potential benefits in reducing the neurodegenerative effects observed in diabetes. For this study, the authors used a rat (male, Wistar) type 2 diabetes model (10% fructose diet for 2 weeks followed by i.p. administration of 40 mg/kg b.w. streptozotocin after an overnight fast) to which they administered Raffia palm wine (150 and 300 mg/kg b.w.) for a period of 5 weeks. Treatment significantly influenced brain oxidative stress by elevating GSH (*p* < 0.05) level, increasing SOD (*p* < 0.05) and CAT (*p* < 0.05) activity. The treatment also decreased Nrf2 expression (*p* < 0.05), AChE activity (*p* < 0.05), and MDA (*p* < 0.05) levels while improving neuronal integrity and reducing heavy metal burden in the brain.

Although the consumption of alcoholic beverages for their benefic effects is a much-disputed topic in the medical community, from a research point of view, the possibilities that arise from fermented beverages and their phytochemicals should always be considered and explored for their application into adjuvant treatments. As the above studies have exemplified, the body of evidence for the neuroprotective effects of these phytochemicals is important and warrants further investigation for alternative applications in food items that do not have unwarranted health side effects.

### 3.5. Tea Phytochemicals

Brewing and consuming tea has been an integral part of human diet culture for millennia. Tea has been classified into four major categories: black, green, Pu-er, and oolong, according to the degree of fermentation that the tea leaves undergo during processing [92]. The typical components of fresh tea leaves (in decreasing order of dry weight %) include flavonols, proteins (including enzymes such as peroxidases and polyphenol oxidase; amino acids such as l-theanine, etc.), mono and polysaccharides, cellulose, lignin, phenolic acids, depsides, caffeine, lipids, chlorophyll (and other pigments), theobromine, and other volatiles [93]. The creation of black tea through processing and fermentation induces catechin oxidation with the formation of thearubigins, theaflavin, and theasinensin A and D [94]. The high phytochemical contents of black tea make it an ideal candidate as a functional food in AD. To this end, Mathiyazahan et al. (2015) [45] investigated the neuroprotective effects of black tea (*Camellia sinensis* (L.) Kuntze) on a rat model (male Albino, 10–12 weeks old) of aluminum chloride-induced AD (i.p. injection, 100 mg/kg b.w./day for 60 days). The tea (0.75%, 1.5%, and 3% concentration) was brewed fresh every day and given to the animals for ad libitum consumption in drinking water over the entire 60 days period in which the AlCl3 injections were administered. The 1.5% dark tea treatment incurred a significant change in AChE activity (marked decrease, *p* < 0.05) and oxidative stress markers, increasing AlCl3-reduced activity of CAT (*p* < 0.05), SOD (*p* < 0.05), TBARS (*p* < 0.05), GPx (*p* < 0.05), and GSH levels (*p* < 0.05) in the hippocampus and cortex of rats. Likewise, this tea concentration treatment ameliorated the AlCl3-induced protein expression changes in apoptotic indices, decreasing the expression of cytochrome c (mitochondrial fraction) (*p* < 0.05), Bax (*p* < 0.05), caspase 3 (*p* < 0.05), 8 (*p* < 0.05), and 9 (*p* < 0.05), while increasing Bcl-2 expression (*p* < 0.05) and cytochrome c expression (cytosol fraction) (*p* < 0.05) in the hippocampus and cortex. The AlCl3 treatment induced AD-like pathological changes also through elevation of APP, Aβ1–42, β, and γ secretases protein expression, all of which were significantly inhibited (*p* < 0.05) by the 1.5% dark tea treatment. The changes were reflected in the cognitive function of rats examined using the MWM and the PAT, where an amelioration in cognitive dysfunction was noted in the groups that received 1.5% dark tea treatment.

A second type of tea investigated by Jeong et al. (2020) [46] is Pu’er, a post-fermented tea made from *Camellia sinensis* (L.) Kuntze leaves. The authors explored the tea’s ability to affect cognitive impairment and neuroinflammation in a mouse model (ICR mice, 8 weeks old) of LPS-induced neuroinflammation. The tea was administered daily, orally, in doses of 150 and 300 mg/kg b.w., for a period of 4 weeks. In the final week of the treatment, *Escherichia coli* O55:B5 LPS (400 μg/kg b.w.) was administered once daily by injection. The tea treatment successfully attenuated many of the detrimental effects caused by the LPS administration. The authors reported that Pu’er tea reduced the loss of Nissl substance (*p* < 0.05 for both doses) in the hippocampus as well as the number of GFAP and Iba-1 reactive cells in the hippocampus (*p* < 0.001 for both doses; and *p* < 0.05 for both doses, respectively) and cortex (*p* < 0.01 for both doses; and *p* < 0.01 for the 300 mg/kg b.w. dose, respectively). The tea treatment also successfully lowered LPS-increased brain mRNA expression levels of TNF-α (*p* < 0.001 for both doses), IL-1β (*p* < 0.001 high dose), inducible nitric oxide synthase iNOS (*p* < 0.001 high dose), COX-2 (*p* < 0.001 for both doses). Furthermore, the treatment also influenced NF-κB and MAPK pathways via phosphorylation, as a significant decrease in P-ERK/ERK (*p* < 0.001 for both doses), P-p38/p38 (*p* < 0.01 for both doses), and P-JNK/JNK (*p* < 0.001 for both doses) were noted. Lastly, decreases in BACE-1 (*p* < 0.05 high dose), iNOS (*p* < 0.01 for both doses), and COX-2 (*p* < 0.01 high dose) protein expression were reported. These results were consistent with findings in the MWM and PAT, where rats treated with both doses of tea presented improved cognitive function, yet the higher dose demonstrated a stronger outcome.

The investigated teas have shown significant anti-inflammatory and antioxidant effects as well as the ability to improve cognitive function in AD animal models. These present results indicate that further investigation into the long-term neuroprotective effects of tea has the potential to uncover a valuable resource in AD adjuvant therapy.

### 3.6. Phytochemicals from Traditional Medicines

Asian countries have a long history of making use of the increased antioxidant and anti-inflammatory properties of plant phytochemicals through the use of traditional herbal medicines. However, only a handful of studies have investigated their effects on AD pathology. The following fermented traditional medicines were investigated for their neuroprotective effects in AD: red mold rice, Sipjeondaebotang, *Ganoderma lucidum*, Gumiganghwal-tang, *Cordyceps cicadae*, and *Codonopsis lanceolata.*

Of these, *Monascus*-fermented red mold rice is a traditional Chinese medicine rich in monacolins that is also often consumed for its hypolipidemic and hypoglycemic properties [95]. Lee et al. (2007) [47] investigated the effect of the red mold rice fermented with *Monascus purpureus* NTU 568 on a rat model (Male Wistar rats) of AD (i.c.v. infusion of Aβ40 for 28 days, total amount administered 4.9–5.5 nmol/234 µL). The fermented treatment was administered simultaneously with the Aβ40 infusion for 27 days, once a day, orally in doses of 151 mg and 755 mg/kg b.w. Both doses of the fermented product significantly inhibited the Aβ-increased AChE activity (*p* < 0.05), Aβ deposition, iNOS expression, MDA content, ROS levels, and increased SOD activity in the hippocampus and cortex of rats. Significant results were obtained during the behavioral tests conducted (MWM and PAT), with both dosages being effective at ameliorating Aβ-induced cognitive deficits. In another study by the same authors [48], the effects of *Monascus*-fermented red mold rice (same dosages and period) on a rat model of AD (i.c.v. infusion of 4.9–5.5 nmol Aβ40 for 28 days) and hyperlipidemia (diet consisting of 72.7% chow, 2.67% butter fat, and 1% cholesterol) was investigated. Interestingly, AD rats fed the hyperlipidemic diet expressed more serious modifications than those on a normal chow diet. The high dose of red mold rice significantly decreased cholesterol levels and Aβ deposition in the hippocampus and cortex. Both doses significantly decreased ROS levels, MDA content, ApoE expression, β-secretase expression, β-secretase activity, and increased sAPPα expression in the hippocampus and cortex of rats. The red mold rice treatment (at both doses) successfully attenuated the AD and hyperlipidemic-induced cognitive deficits, as seen through the MWM and PAT cognitive tests.

Sipjeondaebotang is a traditional Korean medicine composed of an assortment of herbs that have been used for the treatment of fatigue. A study conducted by Park et al. (2016) [49] explored the effects of Sipjeondaebotang and *Lactobacillus*-fermented Sipjeondaebotang on a mouse (C57BL/6 mice, 4 weeks old) model of scopolamine-induced memory impairment (1 mg/kg b.w./day for a total of 21 days). Following the scopolamine injection, the treatments were administered orally, daily, in doses of 125, 250, and 500 mg/kg b.w. Additionally, the authors split their mice group so as to include the possibility of studying proliferation as well as new cell survival. For the proliferation study, mice in each group were administered i.p. injections of 5′-Bromo-2′-deoxyuridine (BrdU, 50 mg/kg b.w. twice daily for 3 days), a marker of proliferative cells in the S-phase on the last three days of scopolamine and Sipjeondaebotang treatment. While for the new cell survival study, the mice received i.p. injections with BrdU for three consecutive days prior to the scopolamine and Sipjeondaebotang treatment. The fermented product led to an increase in survival and proliferation of BrdU-positive cells and immature/mature neurons at all dosages (*p* < 0.05). In contrast, the original product failed to impact hippocampal neurogenesis during the scopolamine treatment significantly. Likewise, only the fermented product normalized the scopolamine-increased activity of AChE (*p* < 0.01), decreased levels of aCh (*p* < 0.05), down-regulated expression of choline acetyltransferase (ChAT) (*p* < 0.05 at 250 mg/kg b.w.; *p* < 0.0001 at 500 mg kg/b.w.). Oxidative stress was also positively influenced by the fermented treatment, with a significant reduction in ROS levels observed (125 mg/kg, 250 mg/kg, 500 mg/kg; *p* < 0.01, *p* < 0.05, *p* < 0.01, respectively). The fermented product was also the only one that attenuated scopolamine-diminished phosphorylated CREB expression and Akt inactivation (*p* < 0.05) in the hippocampus (125 mg/kg, 250 mg/kg, 500 mg/kg; *p* < 0.01, *p* < 0.01, *p* < 0.01, respectively). However, none of the treatments impacted cognitive function (MWM, PAT) or improved hippocampal neurogenesis.

*Ganoderma lucidum* is a traditional medicine whose triterpenoid content has been associated with the ability to increase brain ACh levels and improve memory function in AD [96]. Choi et al. (2015) [50] conducted an investigation into the effects of fermented *Ganoderma lucidum* on a rat model (Sprague-Dawley, male, 6 weeks old) of scopolamine-induced memory impairment (i.p. 1 mg/kg b.w. for 5 days). For a period of 15 days prior to the scopolamine injections, the rats received oral treatments consisting of either *Ganoderma lucidum* aqueous extract, *Ganoderma lucidum* fermented with *B. bifidum,* or *Ganoderma lucidum* fermented with *Lactobacillus sakei* and *Bifidobacterium bifidum* (each in doses of 100 and 300 mg/kg b.w.). The oral treatments continued throughout the scopolamine injections. Following the treatments, the cognitive function of the rats was tested with the aid of the MWM, PAT, rotarod test, and vertical pole test. Significant results were obtained from the double-fermented *Ganoderma lucidum,* where scopolamine-induced memory impairment was attenuated. Motor coordination, on the other hand, was improved for all groups that had received the *Ganoderma lucidum* treatment (fermented and aqueous extract). Additionally, this product was the only one to have significantly lowered AChE activity (*p* < 0.05).

Gumiganghwal-tang is a traditional herbal mix commonly prescribed in East Asian countries for headaches, the common cold, and fever due to its analgesic, neuroprotective, and anti-inflammatory properties [97]. Weon et al. (2016) [51] administered three different doses (50, 100, and 200 mg/kg b.w., orally) of fermented Gumiganghwal-tang and compared its effect with donepezil (1 mg/kg b.w.) on a mouse (ICR mice, male, 4-weeks old) model of scopolamine-induced memory impairment (subcutaneous infusion, 9 min after oral treatments). A dose-dependent effect was noted during behavioral tests, with the mice showing a significant decrease in escape latency in the MWM (*p* < 0.05 in all fermented groups), while in the PAT, only mice that received the highest dose of fermented Gumiganghwal-tang presented a significant increase in shortened latency time (*p* < 0.05). AChE activity in the hippocampus was significantly decreased only in mice that had received the 100 mg and 200 mg/kg b.w. (*p* < 0.05) dosages of Gumiganghwal-tang.

*Cordyceps cicadae*, a fungus that parasitizes *Lepidoptera* larvae, has been consumed as a functional food and used in traditional Chinese medicine for the treatment of palpitations, chronic renal disease, dizziness, and infantile convulsions [98]. Wu et al. (2021) [52] utilized deep ocean water for whole submerged fermentation of *Cordyceps cicada* NTTU 868 prior to administration in an animal model (male, Sprague Dawley, 6–8 weeks old) of AD (continuous 28-day i.c.v. infusion of 180 µL solution containing 24.299 µg Aβ40 and 0.9 mg streptozotocin). The fermented product was administered in the same time frame as the disease-inductive treatments, orally, for a period of 28 days in a dose of 220 mg/kg/day. The effects of ocean water fermented fungus were compared to alternative methods of cultivating the fungus (in ultra-pure water, MgCl2 solution) and N6-(2-hydroxyethyl)-adenosine. The investigated treatment significantly suppressed hippocampal Aβ40 (*p* < 0.05) and BACE levels (*p* < 0.05) while increasing sAPPα (*p* < 0.05). The effects also extended to neuroinflammation were a reduction in the expression of TNFα (*p* < 0.05), IL-6 (*p* < 0.05), and IL-1 (*p* < 0.05) at the hippocampal and neocortical levels together with an increase in hippocampal sRAGE anti-inflammatory factor (*p* < 0.05) was observed. The ocean water fermented fungus significantly increased Mg^2+^ concentration (*p* < 0.05) and MAGT1 expression (*p* < 0.05) in the cortex and hippocampus of rats. The modulation of cell Mg^2+^ through MAGT1 expression has been shown to be intimately tied to the expression of BACE and, inherently, Aβ deposition through cleavage of APP by BACE. These results were correlated to the behavioral tests (MWM and PAT), where the fungal treatments effectively improved the memory and spatial learning of AD rats.

*Codonopsis lanceolata* (Siebold. and Zucc.) Trautv. is a flowering plant native to East Asia that is consumed as a functional food but also used in traditional medicine. In a study by He et al. (2011) [53] following fermentation with *Lactobacillus rhamnosus* and *Bifidobacterium longum* B6 the *Codonopsis lanceolata* (667 mg/kg b.w./day) extract was administered to a mouse model of amnesia (scopolamine-induced, 1 mg/kg b.w. 30 min after the administration of the extracts). While the fermented product displayed increased phenolic content, DPPH scavenging activities, powerful antimicrobial activity, and ability to inhibit the activity of α-glucosidase and tyrosinase (*p* < 0.05), the only effects registered on the animal model were from behavioral tests. The passive avoidance test was the only test administered in which authors reported improvement of scopolamine-induced memory deficits in groups treated with the fermented product. In another study by Weon et al. (2014) [54], *C. lanceolata* (doses of 300, 500, and 800 mg/kg b.w.) fermented with *Bifidobacterium longum* (KACC 20587), *Lactobacillus acidophilus* (KACC 12419), and *Leuconostoc mesenteroides* (KACC 12312) was administered to a mouse (ICR mice, males, 3 weeks old) model of scopolamine-induced memory deficit (subcutaneous injection 1 mg/kg b.w.). The treatments, as well as donepezil (1 mg/kg b.w.), were administered orally 90 min prior to the scopolamine injections. After inducing the disease, behavioral tests (MWM and PAT) were conducted. The highest dose of fermented *C. lanceolata* significantly decreased AChE activity (*p* < 0.001) while decreasing pCREB/CREB ratio (*p* < 0.05). All doses of the fermented product increased brain expression of BDNF (*p* < 0.05) and improved cognitive function as tested in the MWM. In the PAT, significant results were obtained only for the highest dose of fermented product.

As demonstrated by the aforementioned studies, herbal medicines offer significant potential in addressing AD through the modulation of oxidative stress. Since oxidative stress directly contributes to neurodegeneration, the use of herbal antioxidant neuroprotection can be regarded as both a preventive and therapeutic approach. Importantly, scientific evidence has confirmed that the majority of herbal compounds possess a favorable safety profile, affordability, and global accessibility, making them a viable option for significantly reducing the burden of dementia and Alzheimer’s disease. However, their utilization is hindered by a few limitations. Unlike pharmaceutical drugs, herbal medicines do not adhere to the same regulatory standards of purity, potency, and quality control. Consequently, the effectiveness and safety of different herbal preparations can vary significantly [99]. The complex composition of herbal medicines is the primary reason for the limited information regarding their constituents, quality, purity, and stability. Unlike conventional medications, herbal medicines typically consist of multiple constituents, making them an appealing option for multi-phytochemical treatments while also posing challenges in standardization and analysis [100]. As a result, identifying the principal active compound in studies exploring their mechanism of action in AD is often challenging and not regularly addressed in experimental research. The herbal supplement industry lacks the same level of regulation from the FDA as prescription drugs, leading to potential discrepancies between the claims made by herbal supplement manufacturers and the supporting scientific evidence, which may be misleading or inaccurate [101]. Herbal medicines can interact with other medications, and certain herbs may have side effects or be toxic in high doses [102]. To overcome these limitations, it is necessary to conduct more high-quality research and provide education to medical healthcare professionals regarding the appropriate use of herbal medicines.

### 3.7. Fermented Functional Food Phytochemicals

The fermented food products investigated for their neuroprotective effects include black carrots, highbush blueberry vinegar, Chinese date, ginger, fucoidan and carrageenan, Kurozu vinegar, aged garlic, curcumin, and date pits.

Black carrots (*Daucus carota* L.) contain important amounts of carotenoids and anthocyanins that exert potent anti-inflammatory and antioxidant activities with possible utilization in the management of hyperlipidemia and hyperglycemia [103]. Park et al. (2016) [55] administered for a period of 8 weeks 1 g/kg b.w./day black carrots that had been fermented with either *Lactobacillus plantarum* or *Aspergillus oryzae* to a rat model (male, Sprague Dawley) of AD (hippocampal infusion of 3.6 nmol/day Aβ25–35 over a period of 2 weeks) and type 2 diabetes (partial pancreatectomy + high fat diet). Treatment began immediately after the pancreatectomy, with disease induction starting after an additional 2 weeks of treatment. The two fermented products were richer in glycated anthocyanins compared to the original product, and groups treated with the fermented products exhibited significantly less cellular amyloid-β deposition (*p* < 0.05), CREB phosphorylation (*p* < 0.05), and Tau protein phosphorylation (*p* < 0.05) in the hippocampus. Additionally, both treatments improved β-cell proliferation, insulin signaling, energy expenditure (*p* < 0.05), and carbohydrate oxidation (*p* < 0.05), as well as decreasing visceral fat mass (*p* < 0.05). Interestingly the *Lactobacillus plantarum* fermented product was more effective than its counterpart at improving insulin sensitivity and secretion in hyperglycemic states. These groups also expressed improved cognitive function in the MWM and PAT.

Hong et al. (2018) [56] obtained highbush blueberry (*Vaccinium corymbosum* L.) vinegar by two-step fermentation, alcoholic fermentation with *Saccharomyces cerevisiae* KCCM 34709 followed by acid fermentation with *Acetobacter* spp. KCCM 40085. The vinegar was then administered to a mouse model (ICR mice) with amnesia (i.p. 1 mg/kg b.w. scopolamine) for a period of 7 days in a dose of 120 mg/kg b.w. Scopolamine was administered once daily, for 7 days, 30 min prior to training in the PAT and Y-maze test. The effects of the vinegar were compared to those of donepezil (5 mg/kg b.w.) and blueberry extract (120 mg/kg b.w.). The animal groups that received the vinegar presented significantly decreased AChE activity in the cortex (*p* < 0.001) and hippocampus (*p* < 0.001), as well as increased levels of ACh (*p* < 0.001). The treatment influenced oxidative stress, with increased SOD (*p* < 0.01), CAT activity (*p* < 0.05), and decreased MDA levels (*p* < 0.01), as well as the CREB/BDNF pathway by significantly increasing BDNF expression (*p*< 0.05), pCREB (*p* < 0.001) and pAKT (*p* < 0.01) in the hippocampus. Positive effects of the treatment were also noted during the Y-maze and PAT, where the vinegar-treated group exhibited significantly attenuated cognitive deficits.

*Ziziphus jujuba*, also known as the Chinese date, are sweet fruits that have been consumed as functional foods but are also used in traditional Chinese medicine for their sleep-inducing effects. The main biologically active components of the fruit consist of phenols, vitamin C, flavonoids, polysaccharides, and triterpenic acids [104]. Kim et al. (2021) [57] administered for a period of 14 days 200 mg/kg b.w./day of *Ziziphus jujuba* fermented with *Saccharomyces cerevisiae* to a mouse (ICR mice, male, 5 weeks old) model after inducing AD (i.c.v injection of 5 µL Aβ25-35 solution of 5 nM concentration, daily for 5 days). The treatment was evaluated in comparison to the unfermented product. The MWM, NORT, and the T-maze test ascertained that the groups treated with both the fermented and unfermented product displayed improved cognitive function. Differences between treatments were noted in oxidative stress markers, where the groups that received the fermented product displayed significantly suppressed levels of MDA (*p* < 0.05) and NO (*p* < 0.05) in the liver, brain, and kidneys.

Ginger (*Zingiber officinale* Roscoe) is one of the most widely used spices. It is nowadays considered a functional food due to its high contents of bioactive constituents such as terpenes, phenolic compounds (shogaols and gingerols), polysaccharides, and organic acids [105]. Huh et al. (2018) [58] investigated the neuroprotective effects of ginger fermented with *Schizosaccharomyces pombe* in a mouse model of AD. Two separate experimental conditions were used to this end. In the first, the authors administered (orally) 100 mg/kg b.w. fermented ginger to mice (ICR, male, 7 weeks old) 30 min prior to inducing amnesia by intraperitoneal scopolamine injection (1.1 mg/kg b.w.). Behavioral tests (NORT and Y-maze test) were administered 30 min after amnesia induction, and fermented treatment was compared to the non-fermented ginger (100 mg/kg b.w.) and donepezil (2 mg/kg b.w). The fermented product exhibited greater anti-amnesic effects compared to donepezil and the non-fermented product. In the second experiment, mice were subjected to a single hippocampal injection of Aβ1-42 (1 mg mL^−1^ concentration, 3 µL in total) followed immediately by oral administration of increasing dosages of fermented ginger depending on the group (50, 100, 200 mg/kg b.w./day) for a period of 14 days. The treatment decreased neuronal cell loss in the CA3 region in a dose-dependent manner, with the highest decrease being registered at 200 mg/kg b.w. (*p* < 0.001). The effect also extended to the synaptic function of neurons which had been heavily disrupted by the Aβ administration. All doses significantly increased the expression levels of SYN (Synuclein) protein in the CA3 region, while an increase in PSD95 expression level was noted only for the highest dose of fermented ginger (*p* < 0.05). The groups that had received 100 mg and 200 mg/kg b.w. fermented ginger also displayed attenuated cognitive deficits in the NORT and Y-maze tests.

Fucoidan and carrageenan are the two main polysaccharides found in seaweed that display strong biological functions due to their unique structure. Recently, they have garnered attention due to their antioxidant, anticoagulant, anti-inflammatory, and immunoregulatory biologic activity [106]. Zhang et al. (2022) [59] administered fucoidan (brown algae) and carrageenan (red algae) fermented with *Pseudoalteromonas carrageenovora* and *Luteolibacter algae* to a rat model of AD (male, Sprague-Dawley rats). Disease induction was performed by continuous infusion of Aβ25-35 (0.005 mg concentration, 300 µL in total) in the CA1 region of the hippocampus over a period of 3 weeks. By microbial fermentation, the algae were broken down into low-molecular-weight fucoidan and λ-carrageenan and administered to the animals along with a high-fat diet (1% algae content). The treatment was administered for 20 days before Aβ administration and continued for another 22 days after. Both the fermented and unfermented algae treatments successfully lowered Aβ deposition in the hippocampus, with the fermented fucoidan group displaying the greatest reduction compared to the control (*p* < 0.01). A similar outcome was achieved with serum glucose concentrations during the oral glucose tolerance test, where the rats treated with fermented fucoidan had the lowest serum glucose compared to the control (*p* < 0.01). These results were also echoed by hippocampal neurotransmitter expression, where both high and low-molecular-weight fucoidan and λ-carrageenan administration significantly increased the expression of BDNF (*p* < 0.01, *p* < 0.01, *p* < 0.05, *p* < 0.001) and ciliary neurotrophic factor (CNTF) (*p* < 0.05, *p* < 0.001, *p* < 0.05, *p* < 0.01) in the hippocampus. Insulin signaling pathways were also positively affected by the treatments, with significant increases registered for pSTAT/STAT (*p* < 0.05, *p* < 0.001, *p* < 0.05, *p* < 0.05), pAkt/Akt (*p* < 0.01, *p* < 0.001, *p* < 0.01, *p* < 0.01), and pGSK-3β/GSK-3β (*p* < 0.05, *p* < 0.05, *p* < 0.05, *p* < 0.05). An analysis of the intestinal flora of the animals revealed that the induction of AD coupled with the high-fat diet led to an increase in *Clostridium*, *Terrisporobacter,* and *Sporofaciens* species, while low-molecular-weight fucoidan and λ-carrageenan increased the number of *Akkermentia* species. Upon performing the behavioral tests (MWM, Y-maze test, PAT), the authors found that the rats that had received low-molecular-weight fucoidan displayed improved memory function.

Kurozu is a traditional Japanese vinegar obtained from the fermentation of unpolished rice that has been previously investigated for its antioxidant properties [107]. Kanouchi et al. (2016) [60] explored the effects of Kurozu vinegar and Kurozu moromi, the solid residue produced after 1 year of fermentation of Kurozu in earthenware jars, on cognitive dysfunction in senescence-accelerated P8 mice (male, 12 weeks old). Two experiments were conducted. In the first, mice were fed a diet with either 0.25% concentrated Kurozu or 0.5% Kurozu Moromi for a period of 4 weeks, while in the second experiment, mice were fed the same diets but for a total period of 24 weeks. Significant results for the first experimental design were obtained only for the concentrated Kurozu, which showed improved cognitive function in the behavioral test (MWM) and decreased amyloid deposition (*p* < 0.05) and plasma TBARS levels (*p* < 0.05) without any effect on brain TBARS levels. The same results were obtained in the second experimental design. The authors also conducted a DNA microarray analysis of HSPA1A mRNA expression. This protein had been associated with the suppression of protein misfolding and aggregation, and in the current study, its expression has been increased by the administration of concentrated Kurozu (*p* < 0.05).

Aged garlic is obtained by submitting fresh garlic (*Allium sativum* L.) to a cold aging process for a period of 20 months or more. By essentially pickling the garlic in alcohol or purified water, this process has been shown to greatly increase its antioxidant and antiglycation properties [108]. Nillert et al. (2017) [61] inquired into the neuroprotective effects of aged garlic extract (produced through ethanol fermentation for 15 months) on a rat (Wistar, male, adults) model of AD. Disease induction was performed by bilateral ventricular injection of Aβ1-42 (1 µL solution, concentration of 1 µg/µL, single dose) after 55 days of daily treatment administration. The fermented product was administered orally daily in doses of 125, 250, and 500 mg/kg b.w. for a period of 65 days. The authors found that Aβ injections greatly affected microglial activation, with clustering of microglia that presented thick, short processes and significantly increased CD11b immunoreactivity in the hippocampus and cerebral cortex. These modifications were attenuated by the garlic treatment at all doses (125 mg *p* < 0.01; 250 and 500 mg *p* < 0.001). Levels of proinflammatory cytokines TNFα and IL-1β were also modified, with the treatment significantly reducing the up-regulation of IL-1β in the hippocampus at all doses (*p* < 0.01). The authors also conducted a NORT in which groups that had received the two highest doses presented an amelioration in their cognitive deficit.

*Cucurma longa* L. is a popular dietary plant with a high phenolic content that presents numerous biological activities, including anti-inflammatory, antioxidant, antimicrobial, and anti-clotting properties [109]. L. Eun et al. (2017) [62] investigated the effect of *Curcuma longa* L. fermented with 5% *Lactobacillus plantarum* K154 on scopolamine-induced memory deficits (i.p. injection of 1 mg/kg b.w.) in mice (ICR, male). The fermented product was administered orally in doses of 50, 100, and 200 mg/kg b.w. 1 h before conducting the acquisition trial in the PAT and 1 h before the first trial session in the MWM (four consecutive days). Scopolamine or donepezil was administered 30 min after the fermented treatment. The mice in the fermented treatment groups performed better in the behavioral tests, with results close to the positive control. Additionally, the groups that received 200 mg/kg b.w. fermented product presented significantly higher expression of BDNF and pCREB in the hippocampus (*p* < 0.05).

Dates are a popular food item enjoyed all over the world. However, the date palm industry has not taken advantage of the massive amounts of seeds that it also generates. With an estimated annual production of 1 million tons of seeds, they can be repurposed and used for their phytochemical compounds in medicine, food, and even energy production [110]. To this end, Saleh et al. (2021) [63] explored the protective effects of date palm pits (*Phoenix dactylifera* L.) fermented with *Trichoderma reesei* on an animal model (Sprague–Dawley male rats) of scopolamine-induced (i.p. 2 mg/kg b.w.) cognitive impairment. The rats received daily oral treatment of either date pit extract (100 mg/kg b.w.), fermented date pit extract (100 mg/kg b.w.), or donepezil (2.25 mg/kg b.w.) for 28 days. The scopolamine injections were administered daily during the last 14 days of the treatments. Results showed that both date pits and the fermented product significantly increased (p < 0.05) the GSH and GST level, SOD, and GPx activities in the brain and serum, while significantly increasing (*p* < 0.05) the levels of NO and TBARS. An effect was noted in lipid contents, where the two treatments lowered (*p* < 0.05) scopolamine-increased levels of triglycerides and total cholesterol in the brain and serum, all while increasing (*p* < 0.05) phospholipid level. These two treatments were also successful in decreasing (*p* < 0.05) Aβ42 level, AChE activity and expression level, mRNA expression levels of Tau protein, as well as expression levels of TNFα and iNOS in the brain. Plasticity markers were also influenced, with scopolamine-induced downregulation of hippocampal BDNF, CREB, and ADAM17 expression being significantly attenuated (*p* < 0.05). Furthermore, during histological analysis, the authors discovered that the rats that had received the fermented date pits presented with important improvement in neuron morphology, fewer degenerative changes, and almost normal histological structure of the hippocampus. The spatial memory of rats was assessed using the MWM. Both treatments improved scopolamine-induced cognitive decline.

This diverse set of studies highlights the ability to achieve a multi-faceted neuroprotective effect in AD by encouraging the adoption of a diverse and equitable diet. Each food presents unique chemical compounds that display an array of effects. Fermented black carrots, which are rich in carotenoids and anthocyanins, have been shown to significantly reduce cellular amyloid-β deposition and improve cognitive function in AD and type 2 diabetes [55]. Highbush blueberry vinegar, which presents ample amounts of acetic acid, phenolic compounds, and anthocyanins, was found to improve cognitive function and decrease AChE activity while increasing ACh levels [56]. Fermented *Ziziphus jujuba* displayed improved cognitive function and suppressed oxidative stress markers [57]. Ginger fermented with *Schizosaccharomyces pombe,* with its diverse composition encompassing terpenes, shogaols and gingerols, polysaccharides, and organic acids, exhibited a protective effect on the cognitive function in AD [58]. *Cucurma longa* L. and its curcuminoids have been shown to impact spatial memory positively and learning through the modulation of the BDNF-CREB pathway [62]. Aged garlic extract, with its representative organosulfur compounds, attenuated microglial activation and reduced up-regulation of proinflammatory cytokines in AD [61]. Fermented date palm pits, abundant in flavonoids, phenolic acids, and tannins, increased the activity of antioxidant enzymes and levels of GSH and GST in the brain and serum of rats with scopolamine-induced cognitive impairment [63]. These studies not only re-emphasize the importance of active consumption of phenol-rich foods but also the potential of fermentation processes to potentiate the bioavailability and bioactivity of phytochemicals naturally contained within these products.

## 4. Conclusions

The present systematic review highlighted an encouraging number of experimental studies that assessed the effects of fermentation on phytochemical content and outcomes in Alzheimer’s Disease. These studies have shown a glimpse into the ability of fermentation to increase the bioavailability and bioactivity of phytochemicals from food products, thereby encouraging both regular consumption of fermented foods and research aimed at constructing functional foods that can be used to prevent and aid in the treatment of AD. While older studies focused on exploring the effects of traditionally fermented foods, the attention of the research community has shifted towards altering the phytochemical contents and increasing the overall potency of natural products by employing fermentation techniques. This is an encouraging aspect in a world where lifestyle choices are impacting health outcomes at a much higher rate than ever before.

As patient preferences have lately tended to lean towards a natural, proactive, and preventive approach, the production of therapies with the aid of familiar processes, such as the fermentation of natural resources, is more likely to be accepted and adopted by the general population as neuroprotective adjuvant therapy for neurodegenerative disorders. Many studies highlighted the ability to obtain small-molecule phytochemicals through fermentation that are not found in raw products. The combined potency of these phytochemicals has shown the ability to surpass the antioxidant, anti-inflammatory, and neuroprotective effects of individual phytochemicals when administered in pure form.

Among the fermented foods investigated, soy isoflavones obtained through fermentation present the largest body of evidence for significant alteration of phytochemical content and outcomes in animal models of AD. Given their promising effects, there is an opportunity to explore their potential as adjuvant therapy for AD. One innovative approach is through the development of nanoceuticals that incorporate soy isoflavones, which not only offer the most potency but also have the potential to target multiple AD pathogenesis pathways at once. By combining the benefits of fermentation with advanced drug delivery technologies, such nanoceuticals could represent a promising approach for improving the treatment of AD in the future.

Moving forward, these findings offer a strong rationale for advancing soy isoflavones obtained through fermentation to human trials in order to evaluate their safety and efficacy in humans. Although many other herbs and foods presented promising results, more studies are required to ascertain their effects and practicality. However, in order to have a correct outlook on the efficiency of investigated fermented products, the quality of experimental in vivo studies needs to be improved. Many of the studies did not analyze the phytochemical content of the investigated product or compare the original product with the fermented counterpart. Additionally, as the risk of bias assessment highlighted, reporting in current animal studies is incomplete and often poor. Adequate, accurate, and complete descriptions of animal selection and allocation concealment are required to achieve reliable and replicable results.

The majority of studies included in the present systematic review suffered from incomplete or confounding disclosure of animal housing, randomization mechanisms for animal allocation, and blinding of caretakers and outcome assessors from knowing which intervention was administered to animals. Studies also often failed to disclose the number of animals selected for certain outcome assessments, methods for selecting animals, or whether randomization was used. While these problems are not limited to the studies included in the present systematic review, as they are generally present in experimental animal studies, it is important to improve the overall quality of reporting in this category of studies.

## Figures and Tables

**Figure 1 foods-12-02102-f001:**
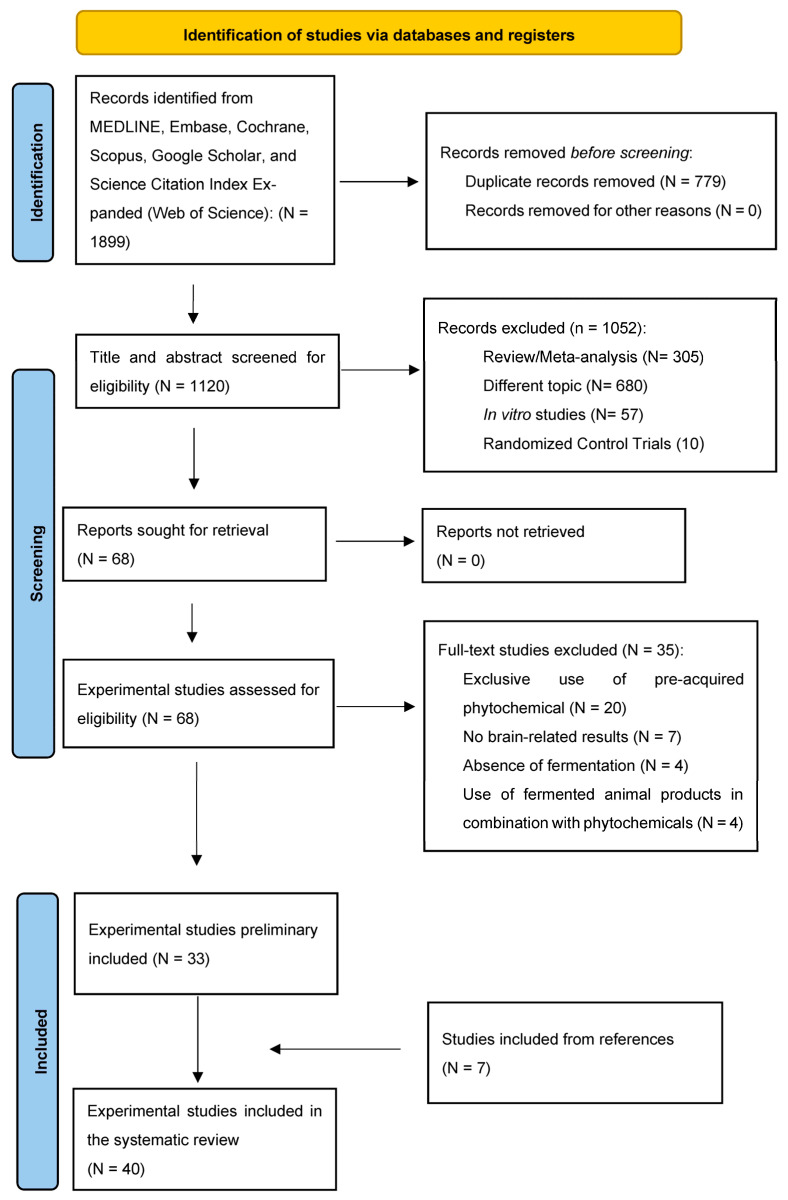
PRISMA flow diagram for the present systematic review.

**Table 1 foods-12-02102-t001:** Characteristics of studies included in the systematic review.

Reference	Fermented Food	Phytochemical Analysis	Comparison between Fermented and Unfermented Products	Experimental Design	Main Findings
[24]	Fermented defatted soybean with *Lactobacillus pentosus* C29	The contents of main constituents in soybeans fermented (2) and nonfermented (1). Content (mg/g). Genistin: 203.6 ± 27(1); 6.6 ± 2.0(2) Genistein: 8.6 ± 0.8(1); 45.9 ± 15.4(2) Daidzin: 185.1 ± 19.1(1); 18.6 ± 3.3(2) Daidzein: 3.5 ± 2.7(1); 34 ± 10.3(2) Soyasaponin Ab: 918.5 ± 106.4(1); 190.6 ± 48.4(2) Soyasaponin I: 1037.2 ± 55.2 (1) 767.5 ± 37.9(2) Soyasapogenol A: 160.9 ± 67.1 (2) Soyasapogenol B: 91.2 ± 41.6 (2)	Yes	Scopolamine-induced memory impairment (intraperitoneal injection 1 mg/kg b.w., 30 min after administration of test agents)	Inhibition of AChE activity. Increased BDNF expression. Improved cognitive function.
[25]	Soybean fermented with *Lactobacillus plantarum* C29	No	Yes	5XFAD transgenic mice	Decreased amyloid-β, β/γ—secretases, caspase-3 expression, and NF-κB activation. Decreased neuron apoptosis and microglia activation. Increased BDNF expression. Improved cognitive function.
[26]	Defatted soybean fermented with *Lactobacillus plantarum* C29	No	Yes	Intraperitoneal injection of LPS (8 μg/kg b.w./day for 10 days)	Increased CREB phosphorylation and BDNF expression. Inhibited NF-κB activation. Improved cognitive function.
[27]	Soymilk fermented with *Lactobacillus plantarum* TWK10	No	Yes	Deoxycorticosterone acetate (subcutaneously 20 mg/kg b.w. 2 times per week over a period of 90 days)	Decreased blood pressure and AChE activity. Improved oxidative stress status. Improved cognitive function.
[28]	Cheonggukjang obtained by soybean fermentation with *Lactobacillus sakei* 383 and *Bacillus subtilis* MC31	Cheonggukjang analysis Total Flavonoids:17.5 mg/g Total polyphenolic content: 37.2 mg/g. Daidzein 0.086 mg/g Genistein 0.030 mg/g	No	Trimethyltin chloride (i.p. 2.5 m/kg b.w., single dose)	Decreased AChE activity and MDA levels. Increase in NGF concentration and activation of the NGF signaling pathway. Increase in SOD activity. Improved cognitive function.
[29]	Chungkookjang	No	No	Tg2576 transgenic mice	Increased NGF levels. No effects on cognitive function.
[30]	Chungkookjang obtained by soybean fermentation with *Bacillus lichenifomis*	Non-fermented, cooked soybeans (1), Traditionally fermented Chungkookjang (2), Chungkookjang obtained by fermentation with *Bacillus lichenifomis* (3) Proline 1.4 ± 1.9 (1) 0.7 ± 1.3 (2) 25.0 ± 5.7 (3). Adenine: 67.9 ± 33.1 (1) 147.0 ± 15.0 (2) 96.6 ± 30.4 (3). Tyrosine: 71.3 ± 16.5 (1) 222.3 ± 38.2 (2) 66.4 ± 8.2 (3). Leucine/isoleucine 121.9 ± 28.2 (1) 188.3 ± 47.9 (2) 502.5 ± 249.6 (3). Phenylalanine: 271.1 ± 90.0 (1) 1429.5 ± 229.9 (2) 2127.2 ± 635.9 (3). Ser-Pro: 0.0 ± 0.0 (1) 0.0 ± 0.0 (2) 153.7 ± 64.2 (3). Val-Glu: 0.0 ± 0.0 (1) 0.0 ± 0.0 (2) 75.0 ± 47.4 (3). Val-Leu: 0.0 ± 0.0 (1) 22.1 ± 9.3 (2) 167.5 ± 13.7 (3). Glu-Phe: 1739.3 ± 106.6 (1) 58.5 ± 13.9 (2) 365.3 ± 22.8 (3). Daidzin: 545.6 ± 95.8 (1) 100.2 ± 17.1 (2) 79.1 ± 7.6 (3). Genistin: 460.1 ± 162.7 (1) 179.1 ± 21.0 (2) 106.1 ± 11.3 (3). Acetylgenistin: 328.8 ± 78.0 (1) 3.4 ± 3.4 (2) 13.8 ± 2.6 (3). Daidzein: 191.6 ± 126.8 (1) 1837.0 ± 111.2 (2) 1074.6 ± 120.2 (3). Genistein: 181.2 ± 113.8 (1) 1633.5 ± 132.6 (2) 891.5 ± 113.0 (3). B soyasaponin Bb’: 37.5 ± 20.1 (1) 53.0 ± 13.3 (2) 26.3 ± 16.0 (3). E soyasaponin: 51.4 ± 40.2 (1) 133.3 ± 50.8 (2) 602.7 ± 190.5 (3). DDMP Soyasaponin βg: 2021 ± 1245 (1) 210 ± 252 (2) 1185 ± 750 (3).	Yes	Hippocampal infusion of Aβ25–35 (3.6 nmol/day for 14 days) and 90 % pancreatectomy	Decreased Amyloidβ accumulation. Improved insulin signaling in the hippocampus. Restored β-cell mass. Improved cognitive function.
[31]	Doenjang	No	Yes	Hyperlipidemic diet (45.2 kcal% fat and 1% cholesterol) for 11 weeks	Decreased neuron cell loss in the hippocampus, oxidative metabolites, Tau hyperphosphorylation. Reduced mRNA expression of oxidative stress and neuroinflammation-related genes. Improved cognitive function.
[32]	Tempeh	No	No	Senescence-accelerated SAMP8 mice	Improved oxidative stress markers in the cortex, hippocampus, and striatum. Increased nuclear factor erythroid 2-related factor 2 (Nrf2) levels. Reduced Amyloidβ levels. Improved cognitive function.
[33]	Tempeh	No	No	Alloxan-induced pre-diabetes (single i.p. 120 mg/kg b.w.)	No significant improvement in blood glucose or cognitive function.
[34]	Solid-state soybean fermentation with *Bacillus subtilis* MTCC 2616	Units expressed per mg fermented soybean powder. Daidzin 50.31 ± 0.4 µg, Genistin 49 ± 0.3 µg, and Glycitin 23.53 ± 0.6 µg. Nattokinase activity 353 ± 2.3 FU g^−1^.	No	I.c.v. colchicine (15 µg/5 µL, single dose)	Increase in AChE activity. Reduced hippocampal activity of GSH, CAT, and SOD. Decreased lipid peroxidation and carbonyl protein levels. Improved cognitive function.
[35]	Ginseng fermented with *Lactobacillus paracasei* A221	Components of ginseng according to fermentation status. Content (%). Fermented ginseng: 0.0 (Rb1); 0.1 (Rb2); 0.5 (Rc); 0.1 (Rd); 0.2 (Rg1); 0.9 (Compound K). Non-Fermented ginseng: 2.0 (Rb1); 1.7 (Rb2); 2.2 (Rc); 1.3(Rd); 0.4 (Rg1); 0 (Compound K).	Yes	I.c.v. infusion of Aβ1-42 (600 pmol/20 µL per day for 7 days) and bilateral electrocauterization of the vertebral arteries	Ameliorated hippocampal neuron loss. Increased Iba-1 and caspase-3 levels. Improved cognitive function.
[36]	Wild ginseng root fermented with *Pediococcus pentosaceus*	Content of ginsenosides in fermented products Rg5 and Rk1: 21.48 and 18.71 mg/g, respectively.	No	Scopolamine-induced memory dysfunction (i.p. 1 mg/kg b.w. 30 min after administration of test agents)	Decreased AChE activity. Increased ACh level. Improved cognitive function.
[37]	Ginseng fermented with *Pediococcus pentosaceus*	Ginsenoside Contents (mg/g). Cultured wild ginseng root: (Rb1) 51.53 ± 1.34; (Rc) 38.16 ± 1.10; (Rb2) 34.36 ± 1.26; (Rb3) 8.10 ± 0.52; (Rd) 55.90± 0.85; (Rg3) N.D.; (Rk1) N.D.; (Rg5) N.D. Total: 188.06± 4.98. Fermented ginseng root: (Rb1) 9.26 ±0.28; (Rc) 4.93± 0.41: (Rb2) 6.36 ± 0.40; (Rb3) 2.83 ± 0.35; (Rd) 11.26 ± 0.56; (Rg3) 44.26 ± 1.02; (Rk1) 15.93 ± 0.32; (Rg5) 23.10 ± 0.59. Total: 117.96 ± 3.38	No	Male mice: scopolamine-induced memory deficit (i.p. 1 mg/kg b.w. 30 min after administration of test agents). Female mice: D-galactose-induced aging (s.c. 100 mg/kg b.w.) and ovariectomy	Decreased AChE activity, increased Ach level in female mice groups. Decreased MDA levels and increased CAT activity in female groups. Improved cognitive function.
[38]	Traditional fermented kimchi	Conducted previously by authors cited in this present article. The content of active compounds per 1 kg of Kimchi: ascorbic acid, HDMPPA, quercitrin, and quercetin were 0.28, 0.04, 0.03, 0.02, and 0.27 g, respectively. Total phenolic contents of Kimchi were 15.75 ± 3.91 mg of GAE/g extract.	Yes	Singular i.c.v. administration of Aβ25-35 (5 nmol/5 μL)	Decrease in BACE, APP, and phosphorylated Tau protein expression level. Decreased protein expression of ER stress markers, proapoptotic molecules, and CHOP. Increased protein expression of anti-apoptotic molecules. Decreased oxidative stress markers.
[39]	Traditional fermented kimchi	Conducted previously by authors cited in this present article. The content of active compounds per 1 kg of Kimchi: ascorbic acid, HDMPPA, quercitrin, and quercetin were 0.28, 0.04, 0.03, 0.02, and 0.27 g, respectively. Total phenolic contents of Kimchi were 15.75 ± 3.91 mg of GAE/g extract.	Yes	Singular i.c.v. administration of Aβ25-35 (5 nmol/5 μL)	Decreased levels of oxidative stress markers. Increased protein expression level of antioxidant enzymes. Decreased protein expression levels of inflammation-related enzymes. Improved cognitive function.
[40]	Red wine Cabernet Sauvignon	Compounds identified in the Cabernet Sauvignon (mg/L). Gallic acid: 8.1. Protocatechuic acid: 0.9. Caffeic acid derivative: 8.5. p-Coumaric acid derivative: 2.4. Gallotannin: 4.9. Catechin: 7.3. Caffeic acid: 6.6. Syringic acid: 4.5. p-Coumaric acid: 3.6. Flavonoid glycoside: 5.8. Flavonoid: 5.1. Resveratrol: 0.2. Ferulic acid derivative: 1.2. Flavonoid aglycone: 2.1	Yes	Tg2576 mice	Decreased Aβ peptide generation. Increased non-amyloidogenic processing of amyloid precursor protein. Improved cognitive function.
[41]	Muscadine wine	Constituent polyphenolic components in Muscadine wine: Gallic acid, Procyanidin, p-Courmaric acid, Ellagitannin, Cinnaminic acid derivative, Resveratrol, Ellagic acid, Flavonoid, Delphindin, Cyanidin, Petunidin, Peonidin, Malvidin. No quantitative data available.	No	Tg2576 mice	Reduced levels of soluble high molecule weight oligomeric Aβ species in the hippocampus and cerebral cortex. Improved cognitive function.
[42]	Wine polyphenolic extract (100 mg/L gallic acid equivalents)	Quantification (μg/mg of freeze-dried PVPP-white wine extract). Gallic acid: 83.06 ± 5.46 3,4-Dihydroxybenzoic acid: 3.19 ± 1.04 2-S-Glutathionyl caftaric acid (GRP): 5.57 ± 0.19 trans-Caftaric acid: 351.75 ± 20.53 Catechin: 17.76 ± 7.71 Hydroxycinnamic acid: 5.59 ± 2.50 Coutaric acid: 28.24 ± 0.37 Chlorogenic acid: 42.40 ± 0.50 Caffeic acid: 10.55 ± 0.82 Catechin derivative: 23.76 ± 9.13 Hydroxycinnamic acid: 9.51 ± 0.15 Resveratrol derivative: 0.47 ± 0.09 Ferulic acid: 2.49 ± 0.20 Resveratrol: 0.73 ± 0.28 Proanthocyanidin: 9.12 ± 0.27 Proanthocyanidin (oligomer of catechin): 280.63 ± 13.66 Ethyl caffeic: 2.97 ± 0.10 Ferulic acid derivative: 2.67 ± 0.05 Total phenolic compounds 880.38 ± 58.68 µg GAE/ mL Hydroxybenzoic acids: 86.25 ± 4.44 Hydroxycinnamic acids: 461.73 ± 24.41 Catechins plus Proanthocyanidins: 331.27 ± 26.06	No	3xTg-AD mice	Increased brain accumulation of hydroxybenzoic acid derivatives and catechins. Modulation of oxidative stress markers. Decreased levels of Aβ1-42 and Aβ1-40 in the brain.
[43]	Beer enriched with *Saccharomyces cerevisiae*	No	Yes	3xTg-AD mice	Decreased Aβ1-42 in the hippocampus and prefrontal cortex. Reduced pro-inflammatory molecules. Increased concentration of anti-inflammatory molecules. Improved cognitive function.
[44]	Raffia Palm (*Raphia hookeri*) wine	No	No	High fructose diet (10% fructose solution) for 2 weeks followed by Streptozotocin-induced type 2 diabetes (single i.p. 40 mg/kg b.w.)	Improved neuronal integrity and reduced heavy metal burden in the brain. Oxidative stress modulation. Decreased AChE activity.
[45]	Black tea (*Camellia sinensis*)	Compositional analysis of black tea extract. Total polyphenols: 442.17 (mg/100 g gallic acid equivalent) Theaflavin: 2.16 (%) Thearubigins: 19.31 (%) Total catechins: 2.04 (%) Caffeine: 1.81 (%) Theanine: 4.1 (mg/100 mL)	No	Chronic AlCl3 administration (i.p. 100 mg/kg b.w./day for 60 days)	Diminished expressions of APP, Aβ1–42, β and γ secretases. Ameliorated protein expression changes in apoptotic indices. Significantly ameliorated oxidative stress by diminishing the lipid peroxidative products and enhancing antioxidant indices. Improved cognitive function.
[46]	Pu’er tea (*Camellia sinensis*)	Catechin and Epicatechin	No	LPS-induced neuroinflammation (400 µg/kg b.w. for 1 week)	Inhibited the expression of amyloid genesis proteins. Inhibited production of inflammatory proteins. Decreased activation of inflammatory pathways. Decreased expression of inflammatory mediator mRNAs in hippocampal tissue. Improved cognitive function.
[47]	*Monascus*-fermented red mold rice	No	No	I.c.v. infusion of Aβ40 (total of 4.9–5.5 nmol/234 µL) for 28 days	Potently reversed increases of AChE activity, ROS, and lipid peroxidation. Decreased total antioxidant status and SOD activity in the brain. Improved cognitive function.
[48]	*Monascus*-fermented red mold rice	No	No	I.c.v. infusion of Aβ40 (total of 4.9–5.5 nmol/234 µL) for 28 days and hyperlipidemic diet (4.85 kcal/g)	Downregulated Aβ40 formation and deposition by suppressing the cholesterol-raised β-secretase activity and apolipoprotein E expression. Mediated proteolytic process of APP toward neuroprotective sAPPR secretion in the hippocampus. Improved cognitive function.
[49]	*Lactobacillus*-fermented Sipjeondaebotang	No	Yes	Scopolamine-induced memory impairment (i.p. 1 mg/kg b.w./day for a total of 21 days)	Improved neurogenesis in the hippocampus. Decreased AChE activity and increased ACh levels. Improved oxidative stress status. Modulation of the cholinergic system and BDNF/CREB/Akt pathway. Improved cognitive function.
[50]	*Ganoderma lucidum* fermented with *Lactobacillus sakei* and *Bifidobacterium bifidum*	No	Yes	Scopolamine-induced memory impairment (i.p. 1 mg/kg b.w. for 5 days)	Decreased AChE activity. Improved cognitive function. Improved motor coordination.
[51]	Fermented Gumiganghwal-tang	No	No	Scopolamine-induced memory impairment (single s.c. 1 mg/kg b. w. 90 min after administration of test agent)	Decreased AChE activity. Improved cognitive function.
[52]	Whole submerged fermentation of *Cordyceps cicada* NTTU 868 with deep ocean water	No	Yes	I.c.v. infusion of 24.299 µg Aβ40 and 0.9 mg streptozotocin (continuous for 28 days)	Suppressed Aβ40, BACE, and expression of pro-inflammatory markers. Increased Mg^2+^ content in the cortex. Increased expression of sRAGE and inhibited release of inflammatory factors by microglia cells. Improved cognitive function.
[53]	*Codonopsis lanceolata* fermented with *Lactobacillus rhamnosus* and *Bifidobacterium longum* B6	The total phenol content of *C. lanceolata* High-pressure extraction and *L. rhamnosus* fermentation: 8.45 mg GAE/g High-pressure extraction and *B*. *logum* fermentation: 8.25 mg GAE/g High-pressure extraction without fermentation: 7.38 mg GAE/g Conventional extraction without fermentation: 6.69 mg GAE/g Flavonoid content Fermented C. lanceolata extracts with B. logum (0.44 mg RE/g) and L. rhamnosus (0.45 mg RE/g) High-pressure extraction and B. logum fermentation contents of hydroxybenzaldehyde, cinnamic acid, and coumaric acid were 222.1, 202.0, and 178.6 μg/g, respectively. The amounts of cinnamic acid for the two fermented products were more than 6x higher than that of the non-fermented product.	Yes	Scopolamine-induced memory impairment (1 mg/kg b.w. 30 min after administration of test agents)	Inhibited α-glucosidase and tyrosinase activities. Improved cognitive function.
[54]	*Codonopis lanceolata* fermented with *Bifidobacterium longum* KACC 20587, *Lactobacillus acidophilus* KACC 12419, and *Leuconostoc mesenteroides* KACC 12312	No	Yes	Scopolamine-induced memory deficit (s.c. 1 mg/kg b.w.)	Significant decrease in AChE activity. Decrease in pCREB/CREB ratio. Increased brain expression of BDNF. Improved cognitive function.
[55]	Black carrots fermented with *Lactobacillus plantarum* and *Aspergillus oryzae*	No	Yes	Hippocampal infusion of Aβ25–35 (3.6 nmol/day for 2 weeks) and type 2 diabetes (partial pancreatectomy + high fat diet)	Suppressed Aβ deposition in the hippocampus. Potentiated insulin signaling. Improved whole body and hepatic insulin resistance, first-phase insulin secretion, and insulin sensitivity in a hyperglycemic state. Improved cognitive function.
[56]	Highbush blueberry (*Vaccinium corymbosum* L.) vinegar obtained by fermentation with *Saccharomyces cerevisiae* KCCM 34709 and *Acetobacter* spp. KCCM 40085	Content (mg/mL). Blueberry extract (1); Blueberry vinegar (2) L-ascorbic acid 1.73 ± 0.03 (1); 0.34 ± 0.03 (2) ellagic acid 0.66 ± 0.04 (1); 0.56 ± 0.04 (2) gallic acid 0.21 ± 0.01 (1); 0.25 ± 0.01 (2) D-catechin 0.41 ± 0.04 (1);1.74 ± 0.04 (2) vanillic acid 2.25 ± 0.15(1); 0.31 ± 0.04 (2) caffeic acid 2.02 ± 0.42 (1); 5.54 ± 0.52 (2) cyanidin chloride 26.34 ± 0.54 (1);28.54 ± 0.54 (2) epicatechin 20.24 ± 0.66 (1); 22.24 ± 0.56 (2) chlorogenic acid 2.43 ± 0.48 (1); 8.68 ± 0.35 (2) myricetin 2.68 ± 0.35 (1); 5.35 ± 0.31 (2) quinic acid 1.35 ± 0.31 (1);6.98 ± 0.34 (2) naringin 1.15 ± 0.43 (1); 6.25 ± 0.43 (2) kaempferol 1.35 ± 0.31 (1); 6.62 ± 0.38 (2) Data represent means ± SD (*n* = 3).	Yes	Scopolamine-induced memory impairment (i.p. 1 mg/kg b.w./day for 7 days)	Activation of BDNF/ CREB/ AKT signaling. Improved cognitive function.
[57]	Zizyphus jujuba fermented with *Saccharomyces cerevisiae*	No	Yes	I.c.v injection of Aβ25-35 (5 nM/5µL) for 5 consecutive days	Suppressed levels of MDA and NO in the liver, brain, and kidneys. Improved cognitive function.
[58]	Ginger fermented with *Schizosaccharomyces pombe*	No	Yes	Single i.c.v Aβ1–42 (1 mg mL^−1^ concentration, 3 µL in total) Scopolamine-induced amnesia (i.p. 1.1 mg kg/b.w. before behavioral test)	Inhibition of neuronal cell loss. Reinstated pre- and postsynaptic protein levels that were decreased by Aβ1–42 plaque-toxicity. Improved cognitive function.
[59]	Fucoidan and carrageenan fermented with *Pseudoalteromonas carrageenovora* and *Luteolibacter algae*	No	No	Aβ25-35 infusion in the CA1 region of the hippocampus for 3 weeks (0.005 mg/300 µL in total)	Potentiated hippocampal insulin signaling and increased the expression of CNTF and BDNF in the hippocampus. Increased insulin signaling. Increased serum acetate concentrations. Increased *Akkermentia* species in the gut microbiome. Improved cognitive function.
[60]	Kurozu vinegar and Kurozu moromi	No	No	Senescence accelerated P8 mice	Increased mRNA of expression anti-misfolding and aggregation proteins. Decreased Aβ deposition and plasma TBARS level.
[61]	Aged garlic	S-allylcysteine: 30.96 mg/g. Allicin: 32 µg/g.	No	Bilateral ventricular injection of Aβ1-42 (single dose of 1 µg/µL)	Reduced microglial activation. Reduced TNFα and IL-1 levels. Significantly improved short-term recognition memory.
[62]	*Curcuma longa* L. fermented with *Lactobacillus plantarum* K154	The amounts of curcumin, Demethoxycurcumin, and Bisdemethoxycurcumin in freeze-dried powder of fermented *Cucurma longa* were 10.37, 1.68, and 2.33 μg/mg, respectively. The total amount of curcuminoids was 1.44% (14.38 μg/mg).	No	Scopolamine-induced memory deficits (i.p. 1 mg/kg b.w.)	Regulation of CREB and BDNF expression. Improved cognitive function.
[63]	Date palm pits (*Phoenix dactylifera*) fermented with *Trichoderma reesei*	Fermented extract contained higher yield (12 g%), amounts of flavonoids (12.9 μg eq/mg extract) and phenolics (367.11 μg eq/mg) than the non-fermented extract, which contained 5.9 and 301.97 μg eq/mg of flavonoids and phenolics, respectively. Fungal degradation resulted in the appearance of 5 new compounds (Pyrogallol, 3-Hydroxytyrosol, Catechol, Cinnamic acid, and Myricetin) that were not present in the date pit extract.	Yes	Scopolamine-induced cognitive impairment (i.p. 2 mg/ kg b.w.)	Decreases in the levels of TBARS and NO in serum and brain. Increases in GSH level and GST, GPx, and SOD activities. Significant reductions in the activity and the expression level of AChE as well as the level of Aβ42. Significant decreases in the mRNA expression levels of Tau protein and inflammatory markers. Significantly restored the expression levels of ADAM17, BDNF, and CREB. Marked improvement of neuron morphology.

Abbreviations used in Table: ACh—Acetylcholine; AChE—Acetylcholinesterase; ADAM17—A disintegrin and metalloprotease 17; Akt—Protein kinase B; ApoE—Apolipoprotein E; APP—Amyloid precursor protein; Aβ—Amyloidβ; Aβ1–42—Amyloidβ1–42; b.w.—Body weight; BACE-1—β-secretase-1; Bax—Bcl-2-associated X protein; Bcl-2—B-cell lymphoma 2; BDNF—Brain-derived neurotrophic factor; CAT—Catalase; CNTF—Ciliary neurotrophic factor; CREB—Cyclic adenosine monophosphate response element binding protein; GAE—Gaellic acid equivalent; GPx—Glutathione peroxidase; GSH—Glutathione; HDMPPA—(3-(4′-hydroxyl-3′,5′-19 dimethoxyphenyl)propionic acid; i.c.v.—intracerebroventricular; i.p.—intraperitoneal; Iba-1—Ionized calcium-binding adaptor molecule-1; IL-1—Interleukin 1; LPS—Lipopolysaccharide; MDA—Malondialdehyde; mRNA—Messenger RNA; NF-κB—Nuclear factor kappa-light-chain-enhancer of activated B cells; NGF—Nerve growth factor; NO—Nitric oxide; Nrf2—Nuclear factor erythroid 2–related factor 2; P-ERK—Phosphorylated extracellular regulated kinase; pAkt—Phosphorylated Protein kinase B; ROS—Reactive oxygen species; sAPPα—Soluble amyloid precursor protein; SOD—Superoxide dismutase; sRAGE—Soluble receptor for advanced glycation end products; TBARS—Thiobarbituric acid reactive substances; TNFα—Tumor necrosis factor α.

**Table 2 foods-12-02102-t002:** Risk of bias results for individual studies (SYRCLE RoB tool [22]).

Study Reference	Selection Bias (Sequence Generation)	Selection Bias (Baseline Characteristics)	Selection Bias (Allocation Concealment)	Performance Bias (Random Housing)	Performance Bias (Blinding)	Detection Bias (Random Outcome Assessment)	Detection Bias (Blinding)	Attrition Bias (Incomplete Outcome Data)	Reporting Bias (Selective Outcome Reporting)
[24]	UNCLEAR	LOW	UNCLEAR	LOW	UNCLEAR	LOW	UNCLEAR	HIGH	LOW
[25]	HIGH	UNCLEAR	UNCLEAR	LOW	UNCLEAR	UNCLEAR	UNCLEAR	HIGH	LOW
[26]	HIGH	LOW	UNCLEAR	LOW	UNCLEAR	UNCLEAR	UNCLEAR	LOW	LOW
[27]	HIGH	LOW	UNCLEAR	LOW	UNCLEAR	UNCLEAR	UNCLEAR	HIGH	LOW
[28]	HIGH	UNCLEAR	UNCLEAR	LOW	HIGH	UNCLEAR	HIGH	HIGH	LOW
[29]	HIGH	UNCLEAR	UNCLEAR	LOW	UNCLEAR	UNCLEAR	UNCLEAR	LOW	LOW
[30]	UNCLEAR	UNCLEAR	UNCLEAR	LOW	UNCLEAR	UNCLEAR	UNCLEAR	HIGH	LOW
[31]	UNCLEAR	UNCLEAR	UNCLEAR	LOW	HIGH	UNCLEAR	UNCLEAR	HIGH	LOW
[32]	UNCLEAR	UNCLEAR	UNCLEAR	LOW	UNCLEAR	UNCLEAR	UNCLEAR	LOW	LOW
[33]	UNCLEAR	UNCLEAR	UNCLEAR	LOW	UNCLEAR	UNCLEAR	UNCLEAR	LOW	LOW
[34]	HIGH	UNCLEAR	HIGH	LOW	HIGH	UNCLEAR	UNCLEAR	LOW	LOW
[35]	HIGH	LOW	UNCLEAR	LOW	UNCLEAR	UNCLEAR	UNCLEAR	HIGH	LOW
[36]	UNCLEAR	UNCLEAR	UNCLEAR	LOW	UNCLEAR	UNCLEAR	UNCLEAR	LOW	LOW
[37]	HIGH	LOW	UNCLEAR	LOW	HIGH	UNCLEAR	UNCLEAR	LOW	LOW
[38]	HIGH	LOW	UNCLEAR	LOW	UNCLEAR	UNCLEAR	UNCLEAR	LOW	LOW
[39]	HIGH	LOW	HIGH	LOW	UNCLEAR	UNCLEAR	HIGH	LOW	LOW
[40]	UNCLEAR	LOW	UNCLEAR	LOW	UNCLEAR	UNCLEAR	UNCLEAR	UNCLEAR	LOW
[41]	UNCLEAR	LOW	UNCLEAR	LOW	UNCLEAR	UNCLEAR	UNCLEAR	UNCLEAR	LOW
[42]	UNCLEAR	LOW	UNCLEAR	LOW	UNCLEAR	UNCLEAR	UNCLEAR	HIGH	LOW
[43]	HIGH	UNCLEAR	UNCLEAR	LOW	UNCLEAR	UNCLEAR	LOW	LOW	LOW
[44]	HIGH	UNCLEAR	HIGH	LOW	UNCLEAR	UNCLEAR	UNCLEAR	LOW	LOW
[45]	UNCLEAR	LOW	UNCLEAR	LOW	UNCLEAR	UNCLEAR	UNCLEAR	LOW	LOW
[46]	UNCLEAR	LOW	UNCLEAR	LOW	UNCLEAR	LOW	UNCLEAR	HIGH	LOW
[47]	UNCLEAR	LOW	UNCLEAR	LOW	UNCLEAR	UNCLEAR	UNCLEAR	LOW	LOW
[48]	UNCLEAR	LOW	UNCLEAR	LOW	UNCLEAR	UNCLEAR	UNCLEAR	LOW	HIGH
[49]	HIGH	LOW	HIGH	LOW	UNCLEAR	UNCLEAR	LOW	HIGH	LOW
[50]	UNCLEAR	LOW	UNCLEAR	LOW	UNCLEAR	UNCLEAR	UNCLEAR	LOW	LOW
[51]	HIGH	LOW	HIGH	LOW	UNCLEAR	UNCLEAR	UNCLEAR	LOW	LOW
[52]	UNCLEAR	UNCLEAR	UNCLEAR	LOW	UNCLEAR	UNCLEAR	UNCLEAR	LOW	LOW
[53]	UNCLEAR	LOW	UNCLEAR	UNCLEAR	UNCLEAR	UNCLEAR	UNCLEAR	LOW	LOW
[54]	HIGH	LOW	UNCLEAR	LOW	UNCLEAR	HIGH	UNCLEAR	HIGH	LOW
[55]	UNCLEAR	LOW	HIGH	LOW	UNCLEAR	UNCLEAR	LOW	HIGH	LOW
[56]	UNCLEAR	UNCLEAR	LOW	LOW	UNCLEAR	UNCLEAR	UNCLEAR	HIGH	LOW
[57]	HIGH	UNCLEAR	UNCLEAR	LOW	UNCLEAR	UNCLEAR	UNCLEAR	LOW	LOW
[58]	UNCLEAR	LOW	LOW	UNCLEAR	UNCLEAR	UNCLEAR	UNCLEAR	LOW	UNCLEAR
[59]	HIGH	HIGH	UNCLEAR	LOW	UNCLEAR	UNCLEAR	LOW	HIGH	LOW
[60]	HIGH	HIGH	UNCLEAR	LOW	HIGH	UNCLEAR	UNCLEAR	HIGH	LOW
[61]	UNCLEAR	LOW	UNCLEAR	LOW	UNCLEAR	UNCLEAR	HIGH	LOW	LOW
[62]	UNCLEAR	LOW	UNCLEAR	LOW	UNCLEAR	UNCLEAR	UNCLEAR	HIGH	LOW
[63]	UNCLEAR	LOW	UNCLEAR	UNCLEAR	UNCLEAR	UNCLEAR	LOW	LOW	LOW

The SYRCLE RoB tool has been used in order to facilitate the critical appraisal of methodology and evidence of the experimental animal studies included in the present systematic review. Judgment of “Low”, “Unclear”, and “High” risk of bias for each item assessed through the SYRCLE tool was established by two independent review authors for each study in part. Disagreements were resolved through consensus-oriented discussion or by consulting a third review author who oversaw the systematic review. The tool included a detailed list of signaling questions for each type of bias (multiple questions per type of bias). When a signaling question was answered with a “no” judgment by the reviewing authors, it was indicative of a High risk of bias for the type of bias analyzed, while a “yes” judgment indicated a Low risk of bias. If there were insufficient details reported in the study to assess the risk of bias properly, a judgment of “Unclear” was assigned. As per the indication of the risk of bias tool, an overall summary score for each study was not performed as it would have required a ranking by importance/weight of each risk of bias domain which is not present in the tool. Hence, individual results for each study are presented in the present table.

## Data Availability

Not applicable.

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
