# Peer review of "Effects of Phytochemicals from Fermented Food Sources in Alzheimer’s Disease In Vivo Experimental Models: A Systematic Review"

_foods, 2023, doi:10.3390/foods12112102_

Round 1

Reviewer 1 Report

The review paper tried to summarize the studies regarding the fermented-food in Alzheimer’s Disease and corresponding chemical explanation. The data collecting method and evidence are scientifically sound. However, some revisions are proposed by the Reviewer.

(1)    At the bottom of “Abstract”, the detailed suggestion should be mentioned to help the future researches. The sentence “Although numerous other phytochemicals from a variety of fermented foods and traditional medicines have been investigated, more research is required in order to verify their effectiveness and practicality” has not substantially meaning for researches.

(2)    Several sentences in “Abstract” are identical with those in “Conclusion”. This can be considered as self-plagiarism. Therefore, some revisions are needed.

Author Response

Point 1: At the bottom of “Abstract”, the detailed suggestion should be mentioned to help the future researches. The sentence “Although numerous other phytochemicals from a variety of fermented foods and traditional medicines have been investigated, more research is required in order to verify their effectiveness and practicality” has not substantially meaning for researches.

Response 1: The text has been changed to include more detailed suggestions, without extending the length of the Abstract too much. The following text has been added in place of the former sentence: “While promising in initial results, other fermented foods and traditional medicines require more detailed research in order to establish their effectiveness and proper utilization. As is, many of the experimental designs lacked phytochemical analysis of the used fermented product or comparison with the non-fermented counterpart. This, coupled with proper reporting in animal studies will significantly raise the quality of performed studies as well as the weight of obtained results.”

Point 2: Several sentences in “Abstract” are identical with those in “Conclusion”. This can be considered as self-plagiarism. Therefore, some revisions are needed.

Response 2: The conclusion section of the Abstract has been re-written so that there is no more overlap with the Conclusion section of the article

Thank you for your positive feedback on our manuscript

Reviewer 2 Report

This is a nice and extensive review paper, which is need to do some small revision. Only one point should be addressed:

- the Authors should mention, which one (or which ones) of the fermented mixture(s) of phytochemicals is (are) the most promissing in terms of application for further trials with human volunteers, before finally being applied as in therapy in the AD.

Author Response

Point 1: - the Authors should mention, which one (or which ones) of the fermented mixture(s) of phytochemicals is (are) the most promissing in terms of application for further trials with human volunteers, before finally being applied as in therapy in the AD.

Response 1: We have added in the Conclusion section of the Article the following text: “Of the fermented foods investigated, soy isoflavones obtained through fermentation present the largest body of evidence for significant alteration of phytochemical content and outcomes in animal models of AD. Given their promising effects, there is an opportunity to explore their potential as adjuvant therapy for AD. One innovative approach is through the development of nanoceuticals that incorporate soy isoflavones, which not only offer the most potency but also have the potential to target multiple AD pathogenesis pathways at once. By combining the benefits of fermentation with advanced drug delivery technologies, such nanoceuticals could represent a promising approach for improving the treatment of AD in the future. Moving forward, these findings offer a strong rationale for advancing soy isoflavones obtained through fermentation to human trials, in order to evaluate their safety and efficacy in humans”

Thank you for your kind review of our systematic review. 

Reviewer 3 Report

Authors send their maniscript entitled "Effects of Phytochemicals from fermented food sources in Alzheimer’s Disease in vivo experimental models: A Systematic Review" for revisoin to the FOODS journal.

Manuscript is well written and description of study has many interesting deatils.

Author Response

Thank you for your kind review of our systematic review. 

Reviewer 4 Report

The authors presented a manuscript entitled “Effects of Phytochemicals from fermented food sources in Alzheimer’s Disease in vivo experimental models: A Systematic Review” in which the phytochemical, from varied sources, such as wine, tea, vegetables and spices, effect on neurodegenerative disease, from many articles was explained and discussed in detail. The review methodology was scientifically sound. The authors followed the PRISMA guidelines and the overall explanation of the review methodology was very clear. The English writing is good. I did not found any major points for revision in this manuscript, my only further suggestion from it is that there was a lack of visual and graphical information. I believe that table or graphics depicting the dose and effect of the discussed phytochemical would improve readability and ease the comprehension of the reader, due to the very extensive nature of the article.

Nevertheless, I found the discussion interesting and the review was well done, warranting from my part no further correction rather than the suggestion cited above.

Author Response

Point 1: The authors presented a manuscript entitled “Effects of Phytochemicals from fermented food sources in Alzheimer’s Disease in vivo experimental models: A Systematic Review” in which the phytochemical, from varied sources, such as wine, tea, vegetables and spices, effect on neurodegenerative disease, from many articles was explained and discussed in detail. The review methodology was scientifically sound. The authors followed the PRISMA guidelines and the overall explanation of the review methodology was very clear. The English writing is good. I did not found any major points for revision in this manuscript, my only further suggestion from it is that there was a lack of visual and graphical information. I believe that table or graphics depicting the dose and effect of the discussed phytochemical would improve readability and ease the comprehension of the reader, due to the very extensive nature of the article.

Response 1: We appreciate the positive feedback on our manuscript and the suggestion to include visual and graphical information. We agree that such information can greatly enhance the readability and comprehension of the article. However, given the large number of investigated fermented foods which naturally contain a large assortment of phytochemicals, as well as the differences in dosages among the studies, it was difficult to create a clear schematic. We did include a comprehensive table in the article that summarizes the different fermented foods and their respective phytochemicals and dosages, as well as their effects on AD in animal models.